# STREAMING-tag system reveals spatiotemporal relationships between transcriptional regulatory factors and transcriptional activity

Hiroaki Ohishi [1], Seiru Shimada[1], Satoshi Uchino[2], Jieru Li [3], Yuko Sato[2,4], Manabu Shintani [1], Hitoshi Owada [1], Yasuyuki Ohkawa [5], Alexandros Pertsinidis [3], Takashi Yamamoto[1], Hiroshi Kimura [2,4] ✉ & Hiroshi Ochiai [1] ✉

Transcription is a dynamic process. To detect the dynamic relationship among protein clusters of RNA polymerase II and coactivators, gene loci, and transcriptional activity, we insert an MS2 repeat, a TetO repeat, and inteins with a selection marker just downstream of the transcription start site. By optimizing the individual elements, we develop the Spliced TetO REpeAt, MS2 repeat, and INtein sandwiched reporter Gene tag (STREAMING-tag) system. Clusters of RNA polymerase II and BRD4 are observed proximal to the transcription start site of *Nanog* when the gene is transcribed in mouse embryonic stem cells. In contrast, clusters of MED19 and MED22 tend to be located near the transcription start site, even without transcription activity. Thus, the STREAMING-tag system reveals the spatiotemporal relationships between transcriptional activity and protein clusters near the gene. This powerful tool is useful for quantitatively understanding transcriptional regulation in living cells.

In multicellular organisms, a specific gene set is expressed in a particular cell type to support cellular functions. Recent studies involving chromosome conformation capture and its derivatives indicate that promoters and distal enhancers interact with each other to activate the expression of specific genes[1]. In particular, genomic regions with multiple enhancers are called super enhancers, which contain multiple binding sites for transcription factors involved in cell identity determination[2–5]. Transcription factors recruit the transcription machinery and coactivators, including the non-phosphorylated form of RNA polymerase II (RNAPII), a Mediator, and the chromatin regulator bromodomain containing 4 (BRD4)[6]. Since these factors form clusters near the transcriptionally active genes[7–13], they may promote the efficient formation of the pre-initiation complex and facilitate the

transcription initiation of genes. During initiation, Ser5 in the YSPTSPS heptapeptide repeat at the C-terminal domain (CTD) of the RNAPII subunit RPB1 is phosphorylated by cyclin-dependent kinase 7 (CDK7)[14]. Subsequently, RNAPII escapes the promoter and is released into productive elongation[6,15].

Many genes are transcribed non-continuously, switching between transcriptionally active and quiescent states, which we here refer to as ON and OFF states, respectively[16]. This is called transcriptional bursting, a universal phenomenon observed in many species and cell types, including mouse embryonic stem cells (mESCs)[16–19]. Although transcription elongation factors are involved in the regulation of transcriptional bursting in a subset of genes, their contribution varies from gene to gene, suggesting that the regulatory mechanism of

[1]Graduate School of Integrated Sciences for Life, Hiroshima University, Higashi-Hiroshima 739-0046, Japan. [2]School of Life Science and Technology, Tokyo Institute of Technology, Yokohama 226-8501, Japan. [3]Structural Biology Program, Memorial Sloan Kettering Cancer Center, New York, NY 10065, USA. [4]Cell Biology Center, Institute of Innovative Research, Tokyo Institute of Technology, Yokohama 226-8503, Japan. [5]Division of Transcriptomics, Medical Institute of Bioregulation, Kyushu University, Fukuoka 812-8582, Japan. ✉e-mail: hkimura@bio.titech.ac.jp; ochiai@hiroshima-u.ac.jp

transcriptional bursting can vary among individual genes[17]. Therefore, to understand the general and gene-specific regulatory mechanisms, further investigation of individual endogenous genes is required.

The MS2 system has often been used as a method to visualize nascent transcripts or transcriptional bursting in living cells[16,20]. The focal accumulation of RNAPII and several regulatory complexes has been detected in the vicinity of pluripotency genes in the ON state in mESCs[8,9,13]. The interaction between enhancers and promoters has also been investigated using MS2 and related systems. The enhancer-gene interaction has shown to be critical in the dynamic regulation of the transcriptional state of certain model reporter genes in *Drosophila* embryos[21,22]. However, their interaction was not associated with the transcriptional bursting of the *Sox2* and *Shh* genes in mESCs[23,24]. Since the MS2 system often inhibits translation when inserted at the 5′ proximal exon of the gene[25], the MS2 repeats are usually inserted downstream of the stop codon in the 3′ untranslated region (UTR). Since the transcription rate is not always constant[26], it is difficult to determine the time delay between the initiation of bursty transcription and the detection of nascent RNA. In addition, the intranuclear gene positions in the OFF state cannot be visualized using only the MS2 system. These technical limitations precluded a more detailed analysis of the dynamics of the clusters involving RNAPII and co-factors in relation to the ON/OFF bursting cycles. Therefore, it is demanding to visualize the gene locus during the transcriptional ON and OFF states in living cells.

We previously developed the Real-time Observation of Localization and EXpression (ROLEX) system to visualize the transcriptional activity and intranuclear localization of a specific endogenous gene by the combined use of MS2 and dCas9 systems[27]. However, MS2 repeats were inserted at the 3′ UTR in the ROLEX system, and somehow the signal-to-noise ratio of dCas9-mNeonGreen at the MS2 repeat DNA was not very high. Hence, in the present study, we develop a novel system for the simultaneous quantification of the nuclear localization and transcriptional activity of the gene regions near the TSS, termed as "Spliced TetO REpeAt, MS2 repeat, and INtein sandwiched reporter Gene tag (STREAMING-tag)" system (Fig. 1a).

## Results
### Basic design of STREAMING-tag system

The STREAMING-tag contains an imaging module comprising MS2 and TetO repeats flanked by a splice donor and a splice acceptor. The imaging module is followed by a G418-selection module containing *IntN-NeoR-IntC*, which encodes an aminoglycoside O-phosphotransferase (APH(3′)), a G418 resistance protein, flanked by split inteins (Fig. 1a). The STREAMING-tag is knocked into a protein-coding region near the TSS. Once the STREAMING-tag is transcribed, the MS2 RNA can be visualized using the MS2 coat protein (MCP) tagged with mScarlet-I, the red fluorescent protein (RFP). The intra-nuclear position of the TetO repeat can be visualized by TetR tagged with mNeonGreen (TetR-mNG). Specifically, we have utilized the Sirius MS2 and optimized TetO repeats, which both use a non-repetitive sequence between the protein binding motifs, unlike the original sequences that consist of simple tandem repeats[28,29]. Thus, genomic instability by repeat-mediated recombination is expected to be minimized. Furthermore, Sirius MS2 has been demonstrated to more effectively amplify dCas9 signals than conventional MS2 when inserted into single guide RNA (sgRNA) in combination with MCP[28]. We here adopted Sirius MS2 for the RNA reporter system. Because the imaging module is flanked by splice donor and splice acceptor, and APH(3′) is flanked by split inteins[29,30], these modules are eliminated by splicing during RNA maturation and protein splicing, respectively. In addition to the amino acids derived from the splice donor and splice acceptor sequences (QG), two amino acids (CF) are added at the N-terminus of C-extein to enhance protein splicing[31], resulting in the insertion of QGCF sequence into the knock-in site of the final protein product. Therefore, the STREAMING-tag-based system is

capable of selecting the knocked-in cells using a selection marker, with minimal effect on the target gene function.

### Optimization of selection marker flanked by split inteins

First, we optimized the *IntN-NeoR-IntC* gene cassette that encodes G418-resistant protein flanked by the split inteins. The functionality of type-I aminoglycoside O-phosphotransferase (APH(3′)-I) from *Klebsiella pneumoniae* sandwiched between split inteins from *Penicillium chrysogenum* PRP8 (PcInt) has already been demonstrated[29]. We compared the *NeoR-I* (encoding APH(3′)-I) and *NeoR-II* (encoding APH(3′)-II, which is 35.1% identical to APH(3′)-I; Supplementary Fig. 1a) sandwiched with either PcInt or a recently reported highly active form of intein Cfa[30]. Histone H2B and mNG fusion proteins with different IntN-NeoR-IntC modules were introduced into mESCs using the piggyBac system. Transfection efficiency was almost the same in different constructs. In contrast, NeoR-I yielded a higher number of G418 resistant cells than NeoR-II, both with PcInt and Cfa (Fig. 1b). Furthermore, Cfa with NeoR-I (CInt-I) was more effective than PcInt with NeoR-I (PInt-I). Next, we analyzed the localization, fluorescence intensity, and size of mNG that was still fused with H2B after intein-mediated protein splicing in cells transfected either with the PInt-I and CInt-I constructs (Fig. 1c). Fluorescence due to mNG was observed in nuclei with both constructs. The average fluorescence intensity was slightly higher in CInt-I (Fig. 1c-e). H2B-mNG bands of the anticipated size were also observed in both constructs, indicating that the inteins were efficiently excised (Fig. 1f). We used the most effective CInt-I for the STREAMING-tag.

### Knock-in of STREAMING-tag into the *Nanog* gene of mESCs

We knocked-in the STREAMING-tag cassette, consisting of splice donor, 24× Sirius MS2, 96× optimized TetO, splice acceptor, and CInt-I into the TSS-proximal coding region of *Nanog* in mESCs (Fig. 2a, Supplementary Fig. 1b,c). Since *Nanog* has multiple transcript variants with distinct TSSs, we selected the knock-in site such that it allows in-frame translation in all the variants; the distances between TSSs and the knock-in site were 311, 314, and 390 bp (Supplementary Fig. 1d). We generated both mono- and bi-allelic *Nanog* STREAMING knock-in cell lines (NSt and bNSt, respectively) (Fig. 2b). In the NSt cells, a six-nucleotide deletion was introduced in the non-knocked-in allele (Supplementary Fig. 1e).

Western blotting analysis showed that NANOG protein levels were slightly and substantially lower in NSt and bNSt cells, respectively, than that in the parental cells (Fig. 2c). Although the *Nanog* gene is important for the maintenance of pluripotency, no obvious abnormalities were observed in the cell morphology of either NSt or bNSt cells. This suggests that even though the level of the NANOG protein decreased with the four amino acid insertion, the protein function may have remained preserved. The NSt cells were designated as *Nanog* STREAMING knock-in cells. These cells were used primarily in the following experiments.

To examine the expression of the *Nanog* mRNA and the splicing of the MS2-TetO cassette, single-molecule fluorescence in situ hybridization (smFISH) was performed on the *Nanog* STREAMING knock-in cells and WT cells using probes designed against the *Nanog* exon region[32] and the TetO repeat region (Fig. 2d). *Nanog* smFISH signals were scattered throughout the cytoplasm, with particularly strong fluorescence spots in the cell nucleus, in both the cell types (Fig. 2d, Supplementary Fig. 1f). *Nanog* smFISH spots with strong fluorescence in cell nuclei are considered to be transcription sites in the ON state[32]. The smFISH signal of TetO was also detected with the strong *Nanog* spots (Fig. 2d, arrowhead), consistent with the splicing out and degradation of the TetO repeat after transcription. Quantification of *Nanog* smFISH spots in *Nanog* STREAMING knock-in and WT cells revealed a slight (25%) decrease in the number of *Nanog* mRNAs in *Nanog* STREAMING knock-in cells compared to that in WT cells

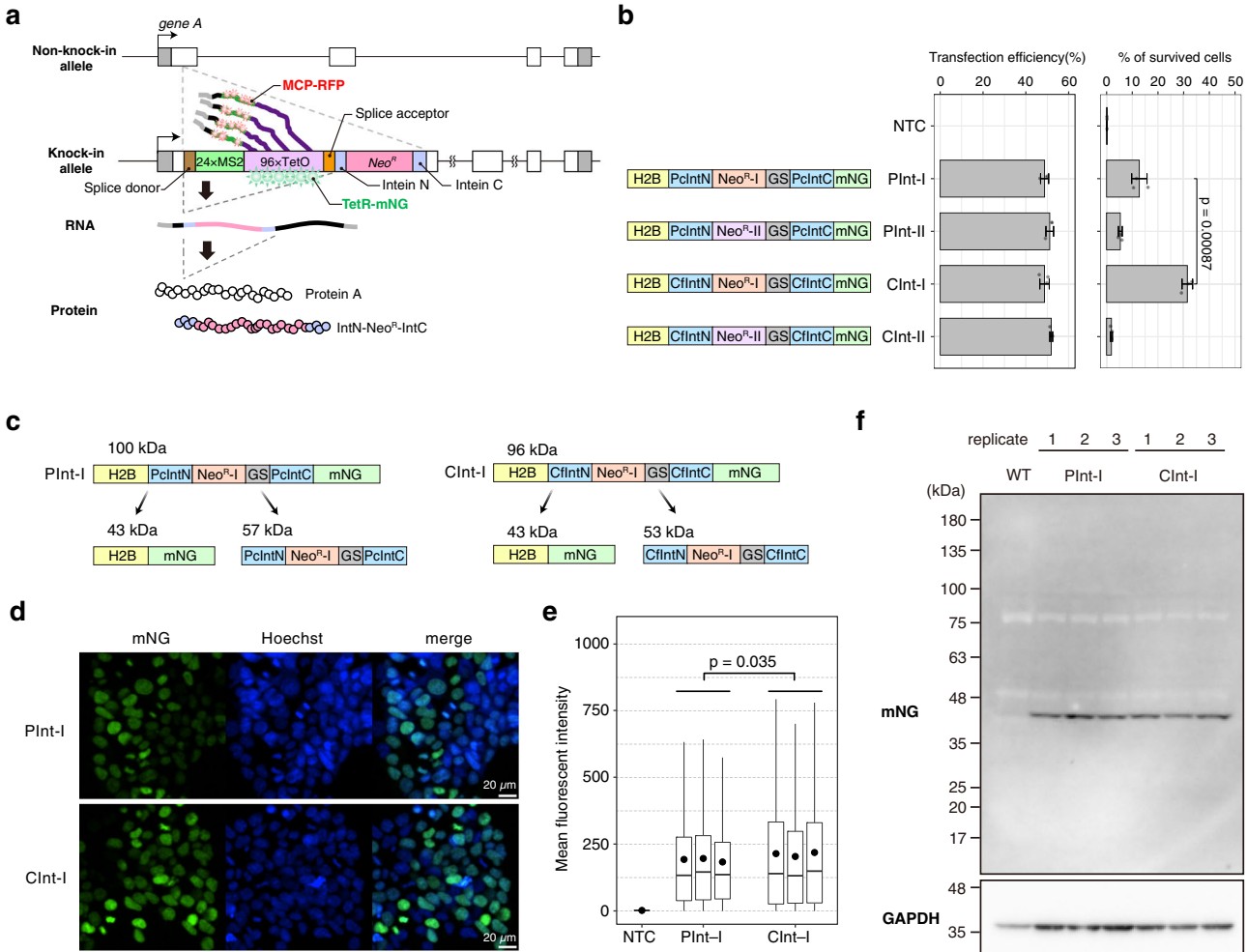

**Fig. 1 | Optimization of selection markers flanked by split inteins. a** Molecular structure and design of the STREAMING-tag. 24×MS2, 24× Sirius MS2 repeat; 96×TetO, 96× optimized TetO repeat. **b** Expression vectors containing split inteins with *Neo^R* that encode G418 resistance protein were introduced into cells. The transfection efficiency and the percentage of cells that survived G418 selection were measured. Data are presented as the means of three biological replicates, and error bars indicate standard deviations. *P*-values correspond to unpaired, two-sided Student's *t*-test. GS, GS linker; NTC, no transfection control. **c** Theoretical structure and size of PInt-I and CInt-I. **d** Fluorescent images of PInt-I and CInt-I cells established after G418 selection in **b**. mNeonGreen (mNG) and Hoechst stained nucleus images (both images are maximum intensity projections of confocal sections) are shown with their merges. Scale bar, 20 μm. **e** Boxplot represents distribution of mean fluorescent intensities of mNG channel in the cell nuclei of NTC, PInt-I and CInt-I are shown. Data for PInt-I and CInt-I include three biological replicates, and circles represent the means of each biological replicate (rep). NTC, *n* = 6,588, Pint-I rep1, *n* = 5,198, Pint-I rep2, *n* = 6,901, Pint-I rep3, *n* = 8,278, Cint-I rep1, *n* = 6,157, Cint-I rep2, *n* = 7,664, Cint-I rep3, *n* = 9,505. Box plots indicate the interquartile range IQR (25–75%) with a line at the median. Whiskers indicate 1.5 times the IQR. *P*-values correspond to unpaired, two-sided Student's *t*-test. **f** Western blot analysis of PInt-I and CInt -I cells using mNG and GAPDH antibodies. Samples for PInt-I and CInt-I include *n* = 3 biological replicates.

(Fig. 2e). The number of nascent RNA molecules in the transcriptional sites was slightly higher (7.3 vs 6.2 on average) for knock-in alleles in *Nanog* STREAMING knock-in cells compared to that in non-knock-in alleles and WT cells (Fig. 2f). By contrast, there was no significant difference in the percentage of cells with transcription sites between WT and *Nanog* STREAMING knock-in cells (Fig. 2g).

We also established mESCs harboring a STREAMING-tag knock-in at *Sox2*, encoding the pluripotency transcription factor SOX2, and *Usp5*, encoding ubiquitin-specific peptidase 5 (Supplementary Figs. 2 and 3). Western blot analysis revealed a moderate decrease in protein expression in monoallelic knock-in cell lines (Supplementary Figs. 2d and 3d). The results of smFISH analysis revealed a slight decrease in RNA expression levels by STREAMING-tag knock-in for both genes (*Sox2*, 9.0%; *Usp5*, 26.9%) (Supplementary Figs. 2e and 3e). The number of RNAs at the transcription site from the knock-in allele was slightly increased in *Sox2* STREAMING-tag knock-in cells (4.9 vs 4.0 on average) but was not changed in *Usp5* STREAMING-tag knock-in cells

(Supplementary Figs. 2f and 3f). As observed for *Nanog*, the percentage of transcriptionally active alleles in both cell lines was not different from WT cells (Supplementary Figs. 2g and 3g). These decreases in RNA expression could be attributed to the tag-altering downstream events, including RNA processing and degradation[33]. Considering the insertion of a significantly long cassette (5.5 kb), another hypothesis is that the decreased expression of the knock-in allele observed via western blotting and smFISH may be explained by the additional time required for elongation and splicing[34–36].

**Visualization of genomic DNA locus by STREAMING-tag system**
To visualize the location and transcription of the STREAMING-tag in the knock-in allele, we expressed wild-type TetR (TetR(WT))-mNG and MCP-RFP in the *Nanog* STREAMING knock-in cell line (Fig. 3a). In the cells where TetR(WT)-mNG was highly expressed, a single mNG spot was clearly detected in the nucleus, whereas MCP-RFP spot was almost undetectable. In contrast, in the cells showing low expression of

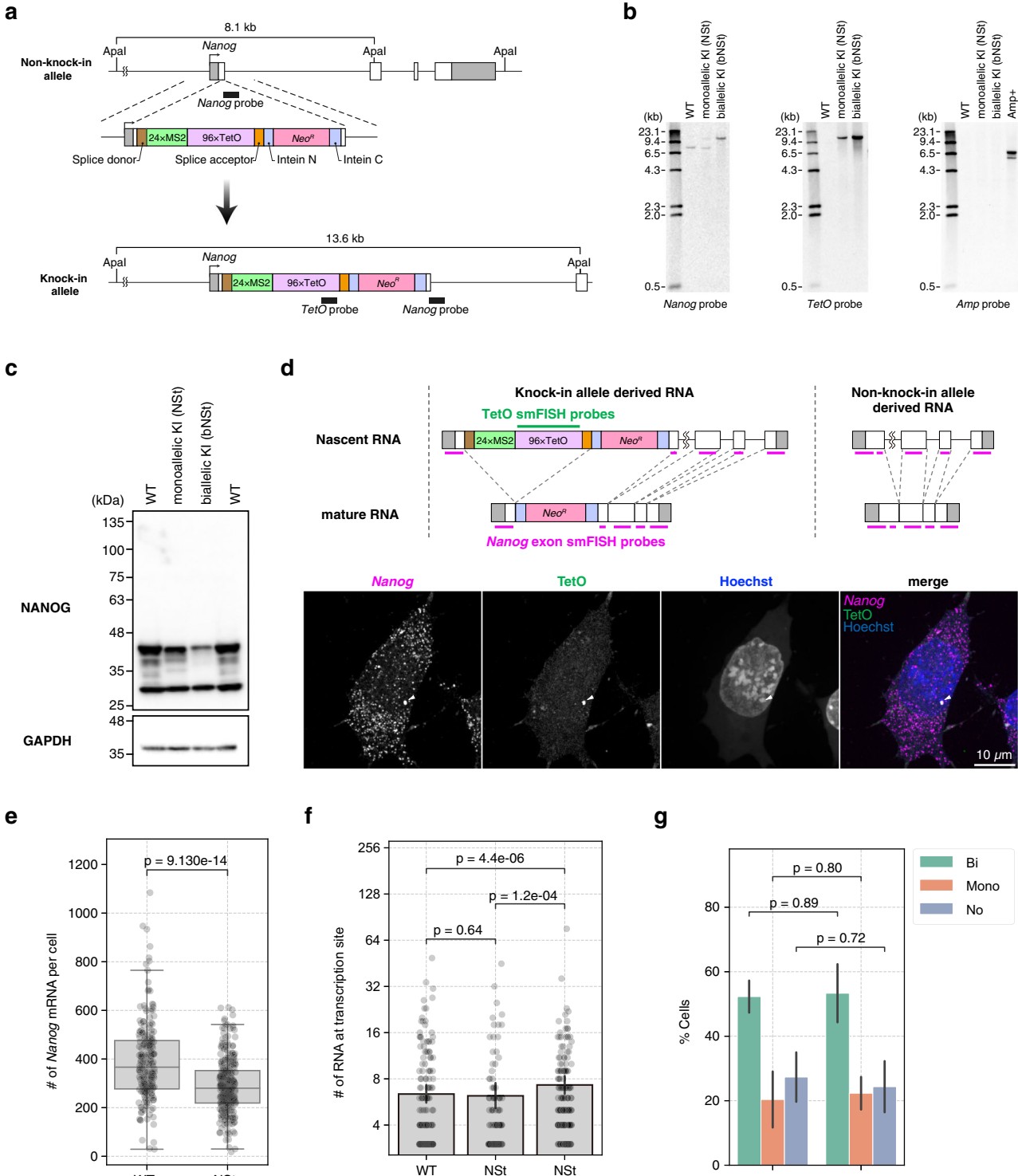

**Fig. 2 | STREAMING-tag knock-in into *Nanog* in mouse embryonic stem cells.**
**a** Mouse *Nanog* gene structure after STREAMING-tag knock-in. **b** Southern blot analysis of mono- and bi-allelic *Nanog*-STREAMING knock-in (NSt and bNSt, respectively), and wild-type (WT) cells. **c** Western blot analysis of NSt, bNSt, and WT cells. **d–g** Single-molecule fluorescence in situ hybridization (smFISH) analysis of NSt and WT cells. **d** Upper panel shows the location of the smFISH probes, and the lower panels show example images of NSt cells. Arrowheads indicate *Nanog* transcriptional spots. Scale bar, 10 μm. **e** Distribution of *Nanog* mRNA counts in WT and NSt cells. WT, *n* = 214 cells; NSt, *n* = 311 cells. *P*-values were determined using two-sided Wilcoxon rank sum test. Box plots indicate the interquartile range IQR

(25–75%) with a line at the median. Whiskers indicate 1.5 times the IQR. **f** Bar graph showing the number of RNA molecules at transcription sites in WT cells (WT), non-knock-in alleles of NSt cells (NSt non-KI), and knock-in alleles of NSt cells (NSt KI); WT, *n* = 203, NSt non-KI, *n* = 110, NSt KI, *n* = 181. *P*-values were determined using two-sided Wilcoxon rank sum test. **g** Bar graph showing the frequency of cells with transcription sites in WT and NSt mESCs. Cells were classified as having no transcription sites (No), only one (Mono), or two (Bi). Data are presented as the means of three biological replicates (more than 25 cells per experiment). The error bars indicate standard deviations. *P*-values correspond to unpaired, two-sided Student's *t*-test.

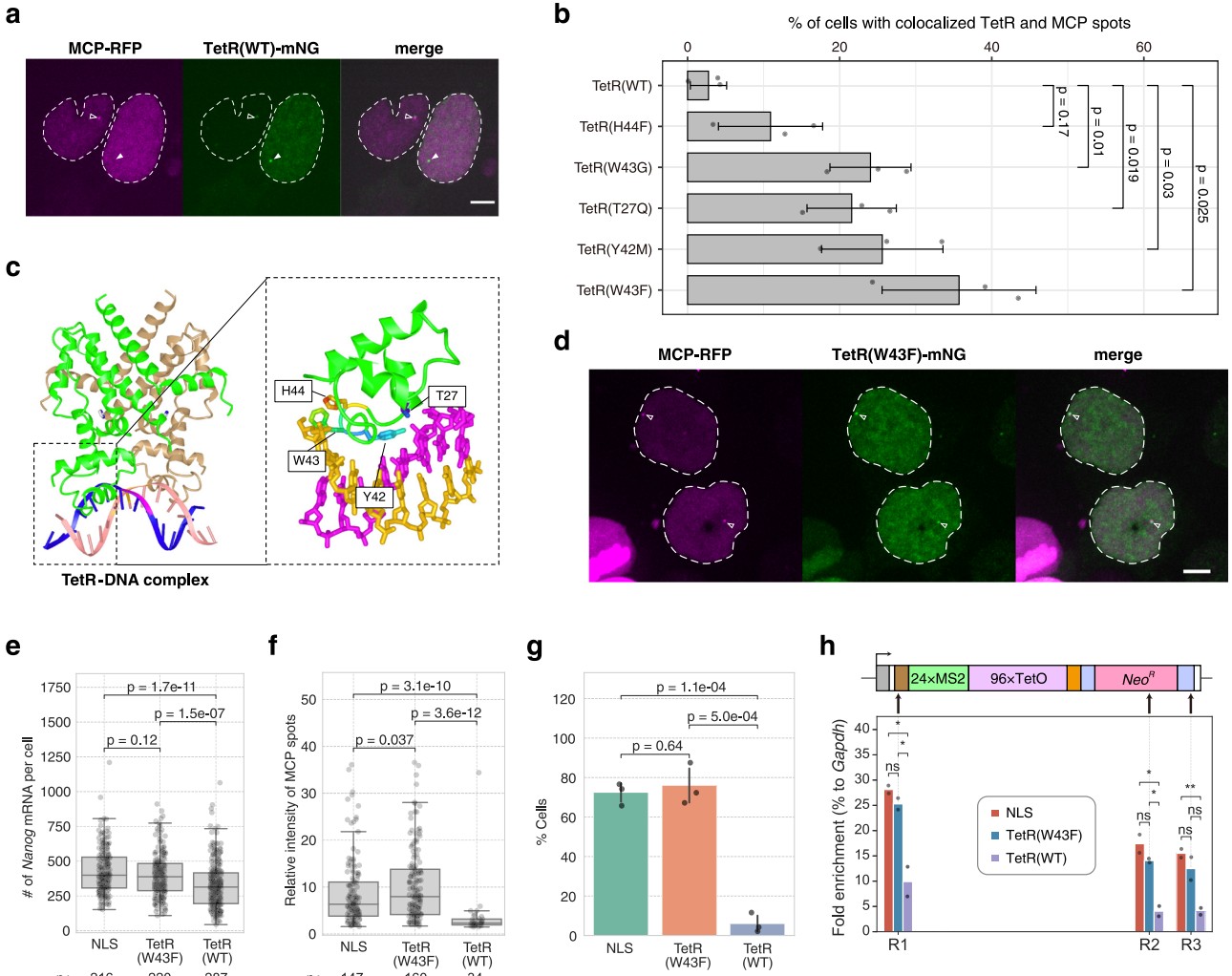

**Fig. 3 | TetR(W43F) is more suitable than TetR(WT) for DNA labeling. a** *Nanog* STREAMING-tag knock-in (NSt) cells transiently expressing TetR(WT)-mNG and MCP-RFP. In cells with high TetR(WT)-mNG expression (right), MCP spots that overlapped with TetR(WT) spots were rarely observed (closed arrowhead). In cells with low TetR(WT)-mNG expression (left), MCP spots tended to overlap with TetR(WT) spots (open arrowhead). Dashed line indicates the cell nucleus. Scale bar, 5 μm. **b** Percentage of cells in which TetR and MCP spots were simultaneously visible and colocalized. Mean values with SD of three biological replicates (>30 cells) are shown. *P*-values correspond to unpaired, two-sided Student's *t*-test. **c** 3D structure of TetR (PDB 1QPI) and mutation site location. **d** Images of NSt cells co-transfected with TetR(W43F)-mNG and MCP-RFP. Open arrowheads indicate TetR(W43F) spots with MCP spots. Dashed line indicates the cell nucleus. Scale bar, 5 μm. **e** Distribution of *Nanog* mRNA counts in NSt derived cell lines expressing MCP-RFP and either NLS-mNG (NLS), TetR(W43F)-mNG (TetR(W43F)), or

TetR(WT)-mNG (TetR(WT)). Box plots indicate the interquartile range IQR (25–75%) with a line at the median. Whiskers indicate 1.5 times the IQR. *n*, number of cells analyzed. *P*-values were determined using two-sided Wilcoxon rank sum test. **f** Distribution of relative fluorescence intensity at MCP transcription sites in cells expressing NLS, TetR(W43F) and TetR(WT). *n*, number of cells analyzed. *P*-values were determined using two-sided Wilcoxon rank sum test. **g** Bar graph showing the percentage of cells with STREAMING-tag knock-in allele in the ON state in NLS, TetR(W43F) and TetR(WT)-expressing cells. Data are presented as the means of three biological replicates (more than 33 cells per experiment). The error bars indicate SD. *P*-values were determined using unpaired, two-sided Student's *t*-test. **h** RNAPII ChIP-qPCR analysis of three locations on STREAMING-tag in NLS, TetR(W43F) and TetR(WT)-expressing cells. Data are presented as the means of *n* = 2 biological replicates.

TetR(WT)-mNG, the MCP-RFP spots were detectable. This suggests that the transcription of the STREAMING-tag was blocked due to the strong DNA binding of TetR(WT). Therefore, we introduced a mutation in TetR to reduce its DNA-binding affinity without affecting the sequence specificity (Fig. 3b, c)[37]. Among the five mutants tested, each with a single amino acid substitution, TetR(W43F) labeling resulted in the detection of both TetR-mNG and MCP-RFP spots in the highest percentage of cells (Fig. 3b, d). To examine the effect of this mutation on the binding property of TetR, we performed fluorescence recovery after photobleaching (FRAP) analysis (Supplementary Fig. 4a, b). The obtained FRAP curve could be well-fitted to a single exponential with the baseline (see Methods). The mobile fraction was not considerably different between TetR(WT) and TetR(W43F) (53.3 ± 11.0% and

49.2 ± 1.4% for TetR(WT) and TetR(W43F), respectively). In contrast, the recovery time constant of TetR(W43F) was substantially smaller than that of TetR(WT) (107.4 ± 34.0 s and 26.3 ± 2.8 s for TetR(WT) and TetR(W43F), respectively). This indicates that TetR(W43F) has a faster dissociation rate than TetR(WT), resulting in a reduced inhibitory effect on transcription (Supplementary Fig. 4a, b).

To confirm whether the TetR(W43F)-mNG and MCP-RFP spots observed using the STREAMING-tag system in the living *Nanog* STREAMING knock-in cells truly represent the *Nanog* locus, we performed DNA-FISH (Supplementary Fig. 4c). *Nanog* STREAMING knock-in cells expressing TetR(W43F) and MCP-RFP were fixed and fluorescence images were acquired. Subsequently, DNA-FISH was performed on the same cells using the *Nanog* probe, which were imaged to detect

DNA-FISH signals that were compared with protein fluorescence. The fluorescent spots of TetR(W43F)-mNG were closely associated with one of the DNA-FISH spots (Supplementary Fig. 4c-e, see Methods). The finding indicates that the knock-in of the STREAMING-tag can visualize specific gene regions in living cells using TetR(W43F)-mNG.

To further validate that TetR(W43F) has a minimal effect on transcription of knock-in allele, we compared smFISH signals using cells expressing NLS-mNG, TetR(W43F)-mNG, and TetR(WT)-mNG as a control of strong binding protein. The number of *Nanog* mRNA detected by smFISH was similar in NLS-mNG and TetR(W43F)-mNG but was reduced in TetR(WT)-mNG (Fig. 3e). In living cells, the fluorescent intensities of MCP transcription spots and the percentage of cells with those spots were also similar in NLS-mNG and TetR(W43F)-mNG, while these were markedly reduced in TetR(WT)-mNG (Fig. 3f, g). Furthermore, chromatin immunoprecipitation followed by qPCR (ChIP-qPCR) using RNAPII-specific antibody revealed that RNAPII occupancy on STREAMING-tag was not affected by TetR(W43F)-mNG like TetR(WT)-mNG (Fig. 3h). These results suggest that TetR(W43F) does not considerably affect STREAMING-tag transcription. TetR(W43F) is hereafter referred to as mutant TetR (mTetR).

## RNA visualization in the STREAMING-tag system

To verify the spots of MCP-RFP represent STREAMING-tag transcripts containing Sirius MS2, we first compared the number of MCP-RFP spots in *Nanog* STREAMING knock-in and WT mESCs. Although the number of spots varied depending on the threshold levels, they were much higher in *Nanog* STREAMING knock-in cells than in WT cells (Supplementary Fig. 4f, g). When the threshold to define spots was set to 3-fold over the average nuclear intensity, 63.5% of *Nanog* STREAMING knock-in cells showed MCP-RFP spots, while spots were not observed in 94.1% of WT cells (Supplementary Fig. 4f, g). We next examined if MCP S47R and R49H mutants, which have reduced MS2 binding ability[38,39], impaired their accumulation at Sirius MS2. The fluorescence intensities of MCP(S47R)-RFP and MCP(S47R, R49H)-RFP on mTetR-mNG spots were much lower than that of WT MCP-RFP (Supplementary Fig. 4h, i). These results support the view that MCP-RFP specifically binds to Sirius MS2 transcribed from STREAMING-tag.

We also estimated the RNA detection sensitivity in the STREAMING-tag system. The smFISH results revealed that an average of 7.3 RNAs are present per transcription site in the STREAMING-tag knock-in allele (Fig. 2f). From live imaging data, the relative intensity of MCP-RFP at the mTetR-mNG spot was ~7-fold higher than the nuclear background intensity (Supplementary Fig. 4i). This finding suggests that MCP-RFP intensity for a single RNA molecule might be at a similar range to that of the nuclear background. As described above, when a threshold of MCP-RFP signal was set to 3-fold over the background, 63.5% of cells showed MCP-RFP spots (Supplementary Fig. 4g). This number is comparable to the percentage of single alleles with transcripts by smFISH (~60%; 50% with biallelic expression, plus half with 20% single allelic expression; Fig. 2g). Since our smFISH analysis defines transcription sites as RNA clusters of three or more molecules detected in the cell nucleus, it is reasonable that the percentage of cells with MCP-RFP spots with relative intensity values exceeding 3 on live imaging is equivalent to that of smFISH. Therefore, detecting a single RNA molecule is not realistic under the experimental conditions used in this study. If multiple RNAs are transcribed simultaneously from an allele, the MCP spot can be detected and judged to be in the ON state. Hence, the ON state can be defined as the state in which the STREAMING-tag knock-in gene is transcribed continuously or bursty, transcribed by multiple RNAPIIs. In addition, for further analysis, to avoid complications due to different cell cycle phases, we excluded cells that showed obvious doublet mTetR signals, which appear after replication of the genome locus during the S and G2 phases[40].

## Verification of the versatility of STREAMING-tag

In the ROLEX system, we knocked-in the MS2 repeat immediately downstream of the *Nanog* stop codon to visualize the transcription with MCP-RFP and the MS2 DNA repeat with dCas9-mNG[27]. Compared to the data obtained from dCas9-mNG in the ROLEX system, the ratio of the mean signal intensity of mTetR-mNG spots to the standard deviation of the nuclear area (SNR, see Methods) in the STREAMING-tag system was significantly higher, indicating improved visualization of the gene locus in the STREAMING-tag system (Fig. 4a).

Using the ROLEX system, we previously reported that the mobility of the *Nanog* locus in mESCs, measured for several minutes, significantly increases in the OFF state compared to that in the ON state, whereas that of *Pou5f1* is independent of its transcriptional state[27]. In the present study, we also tested whether similar behaviors are observed in the STREAMING-tag system by measuring the mean squared change in distance using the distance between the center of mass of the cell nucleus and the genome locus to compensate nuclear movements[27,41,42]. The *Nanog* locus showed higher mobility in the OFF state than that in the ON state (Fig. 4b, Supplementary Movie 1). In contrast, the STREAMING-tag knocked-in the *Pou5f1* locus (Fig. 4b,c, Supplementary Fig. 5) showed the negligible difference in mobility between the ON and OFF states. These results were in accordance with previous data[27], suggesting that the dynamic behavior of gene loci in living cells can be quantified using the STREAMING-tag system.

To confirm the versatility of the STREAMING-tag, we further knocked-in the STREAMING-tag into other genes, including *Wnk1* and *Flnc* (Supplementary Fig. 5). We also expressed mTetR-mNG and MCP-RFP in *Sox2* and *Usp5* STREAMING-tag knock-in cells and the transcription and gene loci were clearly observed in all the cell lines (Fig. 4c). Furthermore, in these cells, we observed considerable differences in mobility between the transcriptional ON and OFF states for *Sox2*, but not for *Wnk1*, *Usp5*, and *Flnc* (Fig. 4d-g).

For *Nanog* and *Sox2*, we also analyzed the diffusional motion of the locus by measuring mean squared displacement (MSD) using short time intervals (~30 ms) for a few seconds[43]. For both genes, diffusion coefficients were higher in the OFF state (0.0077 and 0.0059 $\mu m^2$/s for *Nanog* and *Sox2*, respectively) than in the ON state (0.0037 and 0.0030 $\mu m^2$/s for *Nanog* and *Sox2*, respectively) (Supplementary Fig. 6). This result was consistent with the long-term mobility measurement data (Fig. 4b, d), suggesting that *Nanog* and *Sox2* in the OFF state are less constrained than in the ON state. Thus, STREAMING-tag can be used to track the movement of various genes.

## TSS-proximal transcription imaging by STREAMING-tag system

In cells with STREAMING-tag knocked-in near TSSs, the duration between initiation of bursty transcription and the appearance of the MCP-tagged nascent RNA spot is expected to be shorter than in cells with the MS2 repeat sequence knocked-in downstream of the stop codon. To confirm this, we established a cell line (*Nanog*-STREAMING-PP7), in which STREAMING-tags were biallelically knocked into TSS-proximal coding region of *Nanog* and a PP7 repeat whose RNA can be detected using PCP was knocked into a *Nanog* allele at the 3' UTR[32] (Fig. 5a, Supplementary Fig. 7). We then stably expressed MCP-RFP, PCP-HaloTag, and mTetR-mNG in *Nanog*-STREAMING-PP7 cells (Fig. 5b) and simultaneously visualized transcripts with tags near the TSS (by MCP-RFP) and in the 3' UTR (by PCP-HaloTag) in the same cells. In a typical example, MCP-RFP spots preceded PCP-HaloTag spots (Fig. 5c, d). Cross-correlation analysis[44] showed the asymmetry of the cross-correlation with off-center peaks (Fig. 5e). The signal of PCP-HaloTag (corresponding to 3'UTR transcription) was delayed by ~4 min compared to the signal of MCP-RFP (corresponding to transcription of TSS-proximal region) (Fig. 5e). Thus, STREAMING-tag knocked-in near TSSs allows the quantification of transcriptional activity immediately after initiation of bursty transcription.

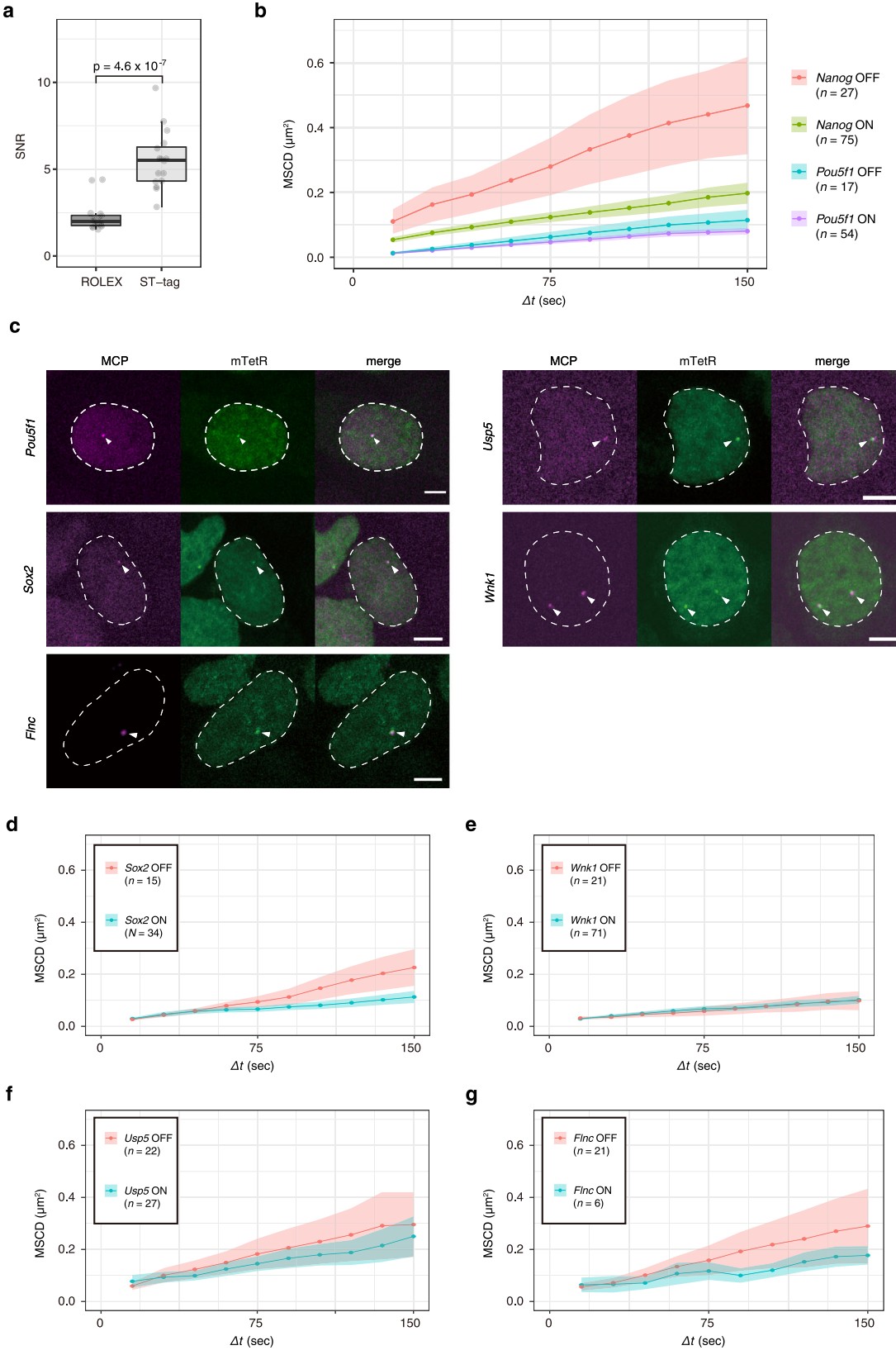

## Protein clusters in close proximity to *Nanog*

We have previously reported that RPB1 and BRD4 form clusters in the vicinity of *Nanog* and *Pou5f1* during the ON state using cell lines in which MS2 repeats were inserted immediately downstream of the stop codon[8,9]. However, it was unclear whether these clusters are also formed in the OFF state (Fig. 6a). Therefore, we established several cell lines in which SNAPtag was knocked into *Rpb1*, *Brd4*, *Med19*, and *Med22* in NSt-GR cells derived from the *Nanog* STREAMING-tag knock-in cell line, which stably express mTetR-mNG and MCP-RFP, using CRISPR-Cas9 gene editing and measured the two-dimensional (2D) distance between mTetR spots and the nearest transcriptional regulatory factor (RF) clusters in the ON and OFF states (Fig. 6b, c, Supplementary

**Fig. 4 | Verification of versatility of the STREAMING-tag. a** Comparison of the signal-to-noise ratio (SNR) of DNA labeling spots between the ROLEX and STREAMING-tag systems. *n* = 16 cells. Box plots indicate the interquartile range IQR (25–75%) with a line at the median. Whiskers indicate 1.5 times the IQR. *P*-value was determined using two-sided Wilcoxon rank sum test. **b** Mean square change in distance (MSCD) of mTetR spots with respect to the center of the nucleus in NSt-NLS-SNAP cells for *Nanog* and PSt-NLS-SNAP cells for *Pou5f1*. The data are classified into the ON and OFF states. Means with standard error of the mean (SEM) are shown. *n*, number of cells analyzed. **c** Imaging of mTetR and MCP spots in knock-in cells with the STREAMING-tag into *Pou5f1*, *Sox2*, *Flnc*, *Usp5*, and *Wnk1*. Dashed lines and arrowheads indicate cell nuclei and mTetR/MCP spots, respectively. Scale bar, 5 µm. In *Wnk1* STREAMING-tag knock-in cells, two spots were observed because the STREAMING-tag was knocked-in into both alleles. **d**–**g** MSCD of mTetR spots with respect to the center of the nucleus in SSt-NLS-SNAP for *Sox2* (**d**), WSt-NLS-SNAP for *Wnk1* (**e**), Ust-NLS-SNAP for *Usp5* (**f**), and FSt-NLS-SNAP cells for *Flnc* (**g**). The data are classified into the ON and OFF states. Means with SEM are shown. *n*, number of cells analyzed.

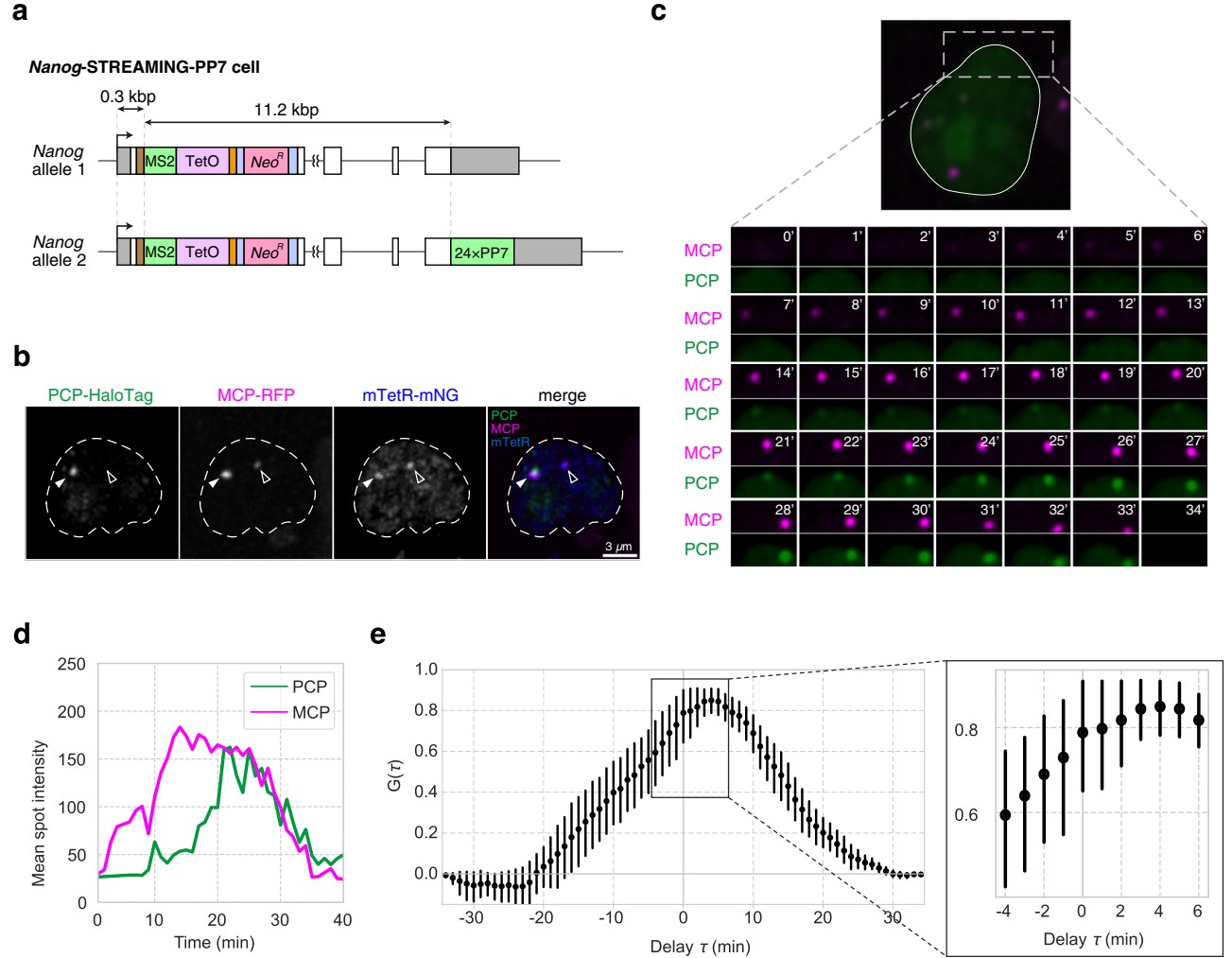

**Fig. 5 | STREAMING-tag system enables quantification of transcriptional dynamics near transcription start site. a** Structure of the *Nanog* gene in a cell line (*Nanog*-STREAMING-PP7) in which the STREAMING-tag is knocked in biallelically near the TSS of *Nanog* and a PP7 repeat is knocked in monoallelically in the *Nanog* 3'UTR. **b** A cell line derived from *Nanog*-STREAMING-PP7 cells and stably expressing MCP-RFP/PCP-HaloTag/mTetR-mNG was established and subjected to live imaging. Representative images of a cell in which MCP-RFP, PCP-HaloTag, and mTetR-mNG spots were observed. Filled white arrowheads indicate locations where MCP-RFP, PCP-HaloTag, and mTetR-mNG spots were observed simultaneously. Unfilled white arrowheads point to locations where MCP-RFP and mTetR-mNG spots were observed simultaneously. Scale bar, 3 µm. **c** Time laps images of MCP-RFP and PCP-HaloTag at 1 min intervals in a cell line used in **b**. **d** Dynamics of the mean fluorescence intensity of the MCP-RFP and PCP-HaloTag at MCP-RFP transcription sites in **c**. **e** Cross-correlation between MCP-RFP and PCP-HaloTag mean spot intensity dynamics. Error bars indicate 95% CI (*n* = 20 cells). The graph on the right shows an enlarged view of the square of the graph on the left.

Fig. 8a-d, Supplementary Data 1). The distances between the mTetR spot and the nearest RPB1 or BRD4 cluster were significantly shorter in the ON state than in the OFF state (median 289 (198) nm and 456 (464) nm for RPB1(BRD4), respectively). However, the distances between mTetR and the nearest MED19 and MED22 clusters were similar to those between mTetR and RPB1 and BRD4 clusters in the ON state (Fig. 6c, d). While the distance to the mTetR spot from the MED19 cluster was not significantly changed in the ON and the OFF states (median 291 nm and 333 nm, respectively), that from the MED22 cluster was slightly shorter in the ON state (median 238 nm and 360 nm, respectively).

Next, we analyzed the correlation between MCP spot fluorescence intensity, as a proxy of transcriptional activity, and the distance between mTetR and the nearest RF clusters. As the median distances of the mTetR-RF clusters are below 350 nm (Fig. 6d) in the ON state, we set a threshold of 350 nm to categorize the distances as short or long. The MCP fluorescence intensity was higher when an RPB1 or BRD4 cluster was within 350 nm of the mTetR spot, while no

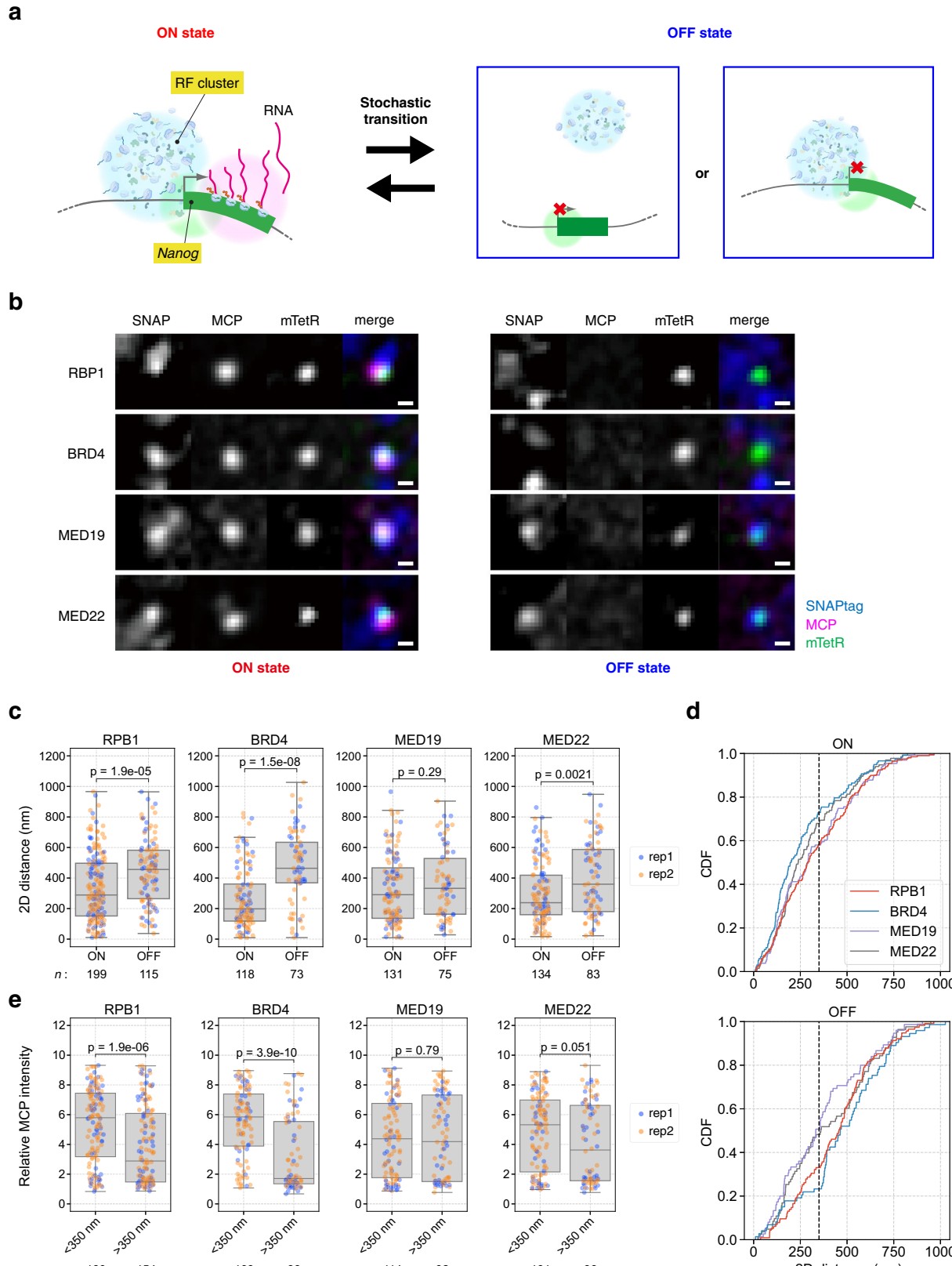

such systematic correlation was observed in case of MED19 and MED22 (Fig. 6e). The fluorescence intensities of the RPB1 and BRD4 clusters nearest mTetR were significantly higher in the ON state than the OFF state, although the difference was modest (RPB1, 24%; BRD4, 17%) (Supplementary Fig. 8e). The intensities of MED19 and MED22 clusters did not show a difference between the ON and OFF states. In

all RFs, sizes of RF clusters in the nearest neighbor of *Nanog* were the same regardless of transcription state (Supplementary Fig. 8f, g). These data suggest that RPB1 and BRD4 clusters are closely associated with *Nanog* during the ON state, whereas MED19 and MED22 clusters are associated with *Nanog* independent of the transcriptional state.

**Fig. 6 | RPB1 and BRD4 form clusters in proximity to *Nanog* only in the ON state in mESCs. a** Hypothetical spatial relationship between transcriptional regulatory factor (RF) clusters and *Nanog*. **b** Relationship between MCP, mTetR, and SNAP-tagged RFs. Using NSt-GR cells, which are derived from the *Nanog* STREAMING-tag knock-in cell line, and expressing mTetR-mNG and MCP-RFP, as a parental cell line, SNAPtag was knocked-in into RF genes. Images with (ON state; left) and without MCP (OFF state; right) are shown. Scale bar, 500 nm. **c** 2D distances between mTetR spots and the nearest SNAPtag clusters in the ON and OFF states. *n*, number of cells analyzed. Box plots indicate the interquartile range IQR (25–75%) with a line at the median. Whiskers indicate 1.5 times the IQR. The blue and orange spots indicate the results of independent experiments. *P*-values were determined using two-sided Wilcoxon rank sum test. **d** Cumulative density function (CDF) of the data in **c**. Black dashed line indicates 350 nm. **e** Distributions of MCP fluorescence intensities at the mTetR spot with the 2D distance between the mTetR spot and nearest SNAPtag cluster at <350 and >350 nm. *n*, number of cells analyzed. Box plots indicate the interquartile range IQR (25–75%) with a line at the median. Whiskers indicate 1.5 times the IQR. The blue and orange spots indicate the results of independent experiments. *P*-values were determined using two-sided Wilcoxon rank sum test.

## Phospho-RNAPII foci form near *Nanog* only in the ON state

To investigate which form of RNAPII is associated with *Nanog* during the ON state, we used genetically encoded modification-specific intracellular antibodies (mintbodies), which consist of a single chain variable fragment (scFv) of a specific antibody and a fluorescent protein[45,46]. In addition to the previously established RNAPII Ser2ph-specific mintbody[47], we generated an RNAPII Ser5ph-specific mintbody (Supplementary Fig. 9, see Methods). We cloned scFv from mouse hybridoma cells producing the RNAPII Ser5ph-specific antibody (Supplementary Fig. 9a) and introduced amino acid substitutions in the framework region to improve the stability (Supplementary Fig. 9b-d). We then confirmed its specificity using enzyme-linked immuno-sorbent assay (ELISA) (Supplementary Fig. 9e,f) and inhibitor treatments (Supplementary Fig. 9g-j).

Cells derived from the *Nanog* STREAMING knock-in cell line and expressing mTetR-mNG, MCP-RFP, and mintbody-SNAPtag were established and imaged following the same procedure as RF cluster imaging. For both Ser5ph and Ser2ph mintbodies, enriched foci were observed close to mTetR and MCP in the ON state (Fig. 7a). We then quantified the distance between the mTetR spots and the nearest mintbody foci (Fig. 7b, c, Supplementary Fig. 8a-c). In case of both RNAPII Ser5ph and Ser2ph, the distance between mTetR and the nearest mintbody foci was significantly shorter in the ON state than that in the OFF state, as observed for RPB1. In addition, the distance from mTetR to the nearest RNAPII Ser5ph foci was similar to that of RPB1 but slightly shorter than that observed for RNAPII Ser2ph foci (Fig. 7b, c). This suggests that the distance of different classes of RNAPII foci from mTetR spot is the following order: the RNAPII Ser5ph foci, the RPB1 cluster potentially including the non-phosphorylated (pre-initiating) or phosphorylated (elongating)-form of RNAPII, and the RNAPII Ser2ph foci. This implies that the STREAMING-tag system can visualize the TSS-proximal region of *Nanog*.

## Discussion

Transcription is a dynamic process that is switched stochastically between the ON and OFF states[16]. We previously showed that transcriptional bursting is controlled by multiple factors, depending on the individual genes[17]. Therefore, it is important to determine the bursting kinetics of individual genes, for which live imaging is one of the best approaches. In this study, we demonstrated that the STREAMING-tag knocked into the TSS-proximal coding region allows the simultaneous determination of nuclear localization and transcriptional state of an endogenous gene without significantly affecting gene function.

Because the MS2 system can inhibit protein translation, MS2 repeats are usually inserted immediately after the stop codon of endogenous genes[25]. However, in this case, we anticipated a time lag between the actual initiation of bursty transcription and the detection of the fluorescent spot by the MS2 system. Since the transcription speed is not constant[26], it is challenging to estimate the precise time points of initiation of bursty transcription, before the fluorescent spot is observed. In contrast to the standard MS2 system, the STREAMING-tag system can be knocked into the TSS-proximal coding region of various genes, such that the gene regions and transcripts can be simultaneously visualized. This TSS-proximal insertion enables the detection of transcripts immediately after the onset of bursty transcription.

In cells with STREAMING-tag and PP7 repeats inserted near the TSS and in the 3′ UTR of *Nanog*, respectively, we found that transcription spots near the TSS were detected approximately 4 min earlier than in the 3′ UTR. The distance between the MS2 repeat within STREAMING-tag and PP7 repeat in 3′UTR was 11.2 kb, and the RNAPII elongation rate was estimated to be approximately 2.8 kb/min, which lies within the characteristic range of RNAPII elongation rate (0.5-4 kb/min)[48] and is comparable to previously reported RNAPII elongation rate in *Nanog* ($2.0 \pm 0.39$ kb/min)[32]. Thus, the STREAMING-tag knock-in near the TSS is potentially useful for observing the transcriptional state immediately after the onset of bursty transcription and consequently in understanding the mechanism of transcriptional regulation. In addition, double knock-in with STREAMING-tag and 3′UTR PP7 can be a useful tool to analyze transcription elongation and processing[50].

Furthermore, the STREAMING-tag system permits the comparison of the localization of RFs involved in transcriptional bursting to the TSS-proximal region (Fig. 7d). In a previous study, the RPB1 cluster was observed to be closer to the MCP transcriptional spot than to the BRD4 cluster in a cell line wherein the MS2 repeats were inserted immediately after the stop codon in the 3′ UTR of *Nanog*[9]. In contrast, in the present study, the BRD4 cluster was closer to the mTetR spot in the ON state than the RPB1 cluster. This difference can be explained by the different insertion sites of the reporter sequences. In the STREAMING-tag system, MCP spots were anticipated to appear soon after the onset of bursty transcription of the *Nanog*. Since the MS2 repeats are possibly co-transcriptionally spliced out through the splice acceptor and splice donor sites within the STREAMING-tag, the MCP spots were observed only when the STREAMING-tag region was continuously transcribed by RNAPII. Therefore, most of the elongating RNAPII complexes may not be in the immediate vicinity of MS2-TetO repeats and mTetR in the STREAMING-tag system (Fig. 7d). As we analyzed the RF cluster nearest to mTetR, RPB1 clusters in the initiation complexes could be primarily detected rather than elongating RPB1 clusters in downstream regions. Thus, a detailed analysis of RF clusters along the gene length may be possible by the combined use of the STREAMING-tag and the standard MS2/PP7 systems to assess the transcriptional activity near the TSS and at the transcription termination site, respectively.

In addition, the tag can be visualized in both the ON and OFF states, while the gene position is detectable only in the ON state when the MS2 system is independently used. In fact, it has been demonstrated using the MS2 system and smFISH that RFs such as RPB1, BRD4, and Mediator form clusters in the nucleus and co-localize with transcriptionally active genes[7-9]. However, the characteristics of the OFF state cannot be analyzed using these systems. Since the formation and dissolution of RF assemblies is anticipated to coordinate with the transcription activity[49], identifying the gene and the associated RFs in the OFF state is crucial to understand the spatial relationship between specific genes and various RF clusters in different states. In this study, we analyzed cells showing singlet spots, likely to be during G1 and S phases before DNA replication. In a previous study, the BRD4 cluster was found between two MCP spots in the *Nanog* 3′UTR[8]. The STREAMING-tag could reveal the location of gene positions (by

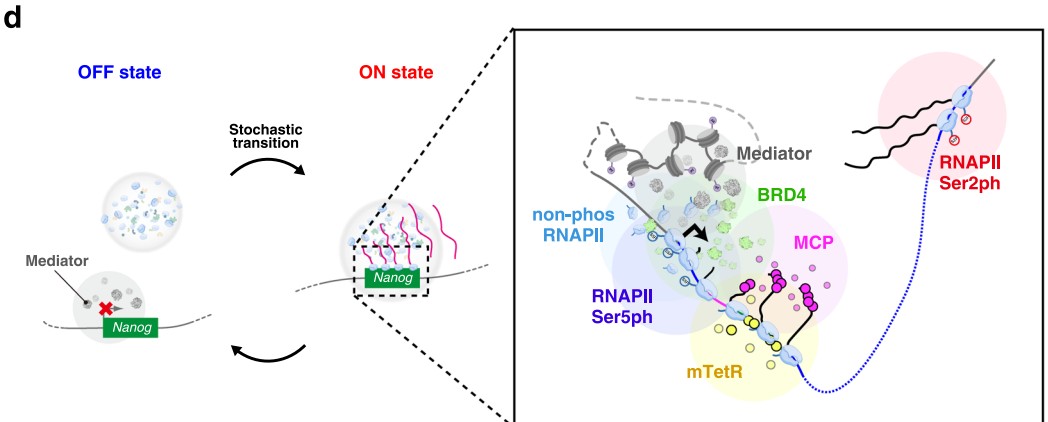

**Fig. 7 | RNAPII Ser5ph and Ser2ph form clusters in proximity to Nanog only in the ON state in mouse embryonic stem cells (mESCs). a** Images of RNAPII Ser5ph- and Ser2ph-mintbodies with MCP and mTetR. Cell lines expressing RNAPII Ser5ph and RNAPII Ser2ph mintbody-SNAPtag were established using NSt-GR as a parental cell line. Scale bar, 500 nm. **b** 2D distance between mTetR spots and the nearest SNAPtag foci in the ON and OFF states. Box plots indicate the interquartile range IQR (25–75%) with a line at the median. Whiskers indicate 1.5 times the IQR. The blue and orange spots indicate the results of independent experiments. RPB1 data are the same as in Fig. 6c, represented for ease of comparison. *n*, number of cells analyzed. *P*-values were determined using two-sided Wilcoxon rank sum test. **c** Cumulative density function (CDF) of the data in **b**. **d** Model of relationship among transcriptional activity, RF clusters, and different forms of phospho-RNAPII clusters in *Nanog* in mESCs.

mTetR) in addition to transcripts (by MCP) and RFs (by SNAP-tagged knock in). As transcription activity can change during the cell cycle, it would be interesting to analyze any difference between different cell cycles (e.g., cells showing singlets vs those showing doublets).

There are some limitations in the STREAMING-tag system. First, knock-in of the STREAMING-tag reduces mRNA expression. The insertion of a 5.5-kb long STREAMING-tag containing one intron could slow down the overall RNA production via transcription elongation and splicing, consistent with the negative correlation between the intron size and gene expression level[34–36]. However, the STREAMING-tag does not appear to significantly affect the fraction of cells with transcription spots, representing the transcription burst frequency[19,50]. It is also possible that the decrease in RNA expression may be due to the addition of the tag-altering mRNA splicing, processing, and export processes, thereby affecting the rate of mRNA degradation. The insertion of the MS2-tag has recently been reported to promote RNA

degradation via the nonsense-mediated mRNA decay (NMD) pathway[33]. Because the STREAMING-tag is removed from mature RNA by splicing, it is unclear if the NMD pathway promotes RNA degradation.

Second, the technical problems associated with the knock-in system also apply to the STREAMING-tag that uses knock-ins. For example, a laborious work would be required for knocking-in to non-expressed genes. Third, since the current system uses the TetO/TetR system, tet-inducible expression system (such as Tet-ON and Tet-OFF)[51] cannot be applied.

In this study, we developed a genetically encoded mintbody that recognizes RNAPII Ser5ph in living cells. As mintbodies repeatedly bind to and dissociate from their respective targets in living cells, they have no effect on cell division, embryo development, and differentiation[52]. The formation of RNAPII Ser5ph-specific mintbody foci was sensitive to the CDK7-specific inhibitor THZ1 (Supplementary Fig. 9g-j), suggesting that the presence of the mintbody does not block Ser5 dephosphorylation. The RNAPII Ser2ph-specific mintbody has recently been developed[47]. Thus, the specific RNAPII forms in the initiation and elongation complexes can now be visualized, which will facilitate future studies regarding transcriptional regulation in living cells.

Using the STREAMING-tag system, we observed distinct association profiles of RF clusters and *Nanog* (mTetR) in the ON and OFF states (Fig. 7d). The ON state of *Nanog* is associated to the nearest RPB1, BRD4 clusters with a proximity <350 nm, whereas the nearest MED19 and MED22 clusters are consistently proximal to *Nanog*, regardless of their transcriptional states. Compared to MED19 clusters that are constantly close to *Nanog*, MED22 clusters showed subtle changes in the distances depending on the transcription states. These findings suggest that individual Mediator subunits interacting with different RFs have different functions and localization in the nucleus[53]. In addition to RPB1, its Ser5- and Ser2-phosphorylated forms were also localized in the vicinity of *Nanog* in the ON state (Fig. 7d). This observation is consistent with the view that transcription complexes containing RNAPII are assembled upon initiation of a transcriptional burst of *Nanog*, rather than significant pools of paused or poised RNAPII being associated with the gene (e.g., *Drosophila* heat-shock, mammalian *β-globin*, *c-myc*, and *c-fos*) during the OFF state[54–56]. The RPB1 spots nearest to mTetR can represent either unphosphoylated, Ser5ph, or Ser2ph forms. However, since only the nearest RPB1 spot was analyzed, most spots were more likely to represent the unphosphorylated or Ser5ph forms than Ser2ph, whose foci are furthest from mTetR. The BRD4 clusters, which are often associated with enhancers were also closer to *Nanog* in the ON state. MED19 and MED22 clusters that are closely located near *Nanog* may serve as a scaffold for the formation of new initiation clusters containing BRD4 and hypophosphorylated RNAPII.

## Methods

### Cell lines

Wild-type (WT) mESCs derived from inbred mice (Bruce 4 C57BL/6 J, male, EMD Millipore, Billerica, MA, USA) and other knock-in derivatives were cultured as described previously[32]. C57BL/6NCr (male) mESCs[17] were used as *Sox2* STREAMING-tag knock-in cells. Briefly, all mESC lines were maintained in 2i medium (Dulbecco's modified Eagle's medium [DMEM, Wako, Osaka, Japan, 197-16275]; 15% fetal bovine serum [FBS, GE Healthcare, Little Chalfont, UK, SH30396.03]); 0.5 mM monothioglycerol solution [Wako, 195-15791]; 1×MEM nonessential amino acids [Wako, 139-15651]; 2 mM L-alanyl-L-glutamine solution [Wako, 016-21841]; 1,000 U/mL leukemia inhibitory factor [Wako, 195-16053]; 20 μg/mL gentamicin [Wako, 078-06061]; 3 μM CHIR99021 [Cayman Chemical, Ann Arbor, MI, USA, 13122]; and 1 μM PD0325901 [Chemscene, Monmouth Junction, NJ, USA, CS-0062]) on a 0.1% gelatin (Sigma-Aldrich, St. Louis, MO, USA, G1890-100G)-coated dish at 37 °C

and 5% CO$_2$. The cell lines used in this study are listed in Supplementary Data 1.

HeLa cells (CCL-2, ATCC) were grown in DMEM high-glucose medium (Nacalai Tesque, Kyoto, Japan) containing 10% FBS (Gibco, Grand Island, NY, USA) and 1% L-glutamine−penicillin−streptomycin solution (GPS; Sigma-Aldrich, St. Louis, MO, USA) at 37 °C in a 5% CO$_2$ atmosphere.

### Plasmids

Plasmids were constructed using common molecular biological techniques. A list of plasmids used can be found in Supplementary Data 2 and is available from Addgene (Watertown, MA, USA; https://www.addgene.org/Hiroshi_Ochiai/).

### Optimization of selection marker flanked by split inteins

C57BL/6 J mESC cell lines ($2.5 \times 10^5$) were plated into each well of a 24-well plate and transfected with 50 ng pCAG-hyPBase[27] and 450 ng Intein-related plasmids (Supplementary Data 1 and 2) using Lipofectamine 3000 (L3000015, Life Technologies, Carlsbad, CA, USA) according to the manufacturer's instructions. After 24 h, the medium was replaced with fresh medium. After another 24 h, the cells were detached using 0.25% trypsin (Thermo Fisher Scientific, Waltham, MA, USA, 15090046) with 1 mM EDTA. Portions of the cells were analyzed using a FACSAria III cell sorter (BD Biosciences, Franklin Lakes, NJ, USA), operated using BD FACSDiva Software (version 8.0.1), to calculate the percentage of mNG-positive cells. Fractions of transfected cells that were not used for flow cytometry analysis were plated into two wells of a 24-well plate, each containing $1.25 \times 10^5$ cells. One well was treated with 200 μg/mL G418 (Geneticin™ Selective Antibiotic, Thermo Fisher Scientific, 10131035). After 2 days, the medium was replaced with fresh medium containing 300 μg/mL G418. After 24 h, the number of surviving cells was counted. Cells expressing PInt-I and CInt-I were passaged and seeded into an 8-well chambered cover glass with #1.5 glass (Cellvis, Sunnyvale, CA, USA, C8-1.5HN). The next day, cells were washed with PBS, fixed with 4% paraformaldehyde in PBS for 10 min, washed twice with PBS, and then treated with PBS containing Hoechst 33342 nucleic acid stain (1:1000) for 10 min. Images were acquired using a Nikon Ti-2 microscope (Nikon, Tokyo, Japan) with a CSU-W1 confocal unit (Yokogawa Electric, Tokyo, Japan), a 20× Nikon Plan Fluor objective lens (NA 0.5), and an iXon Ultra EMCCD camera (Andor Technology, Belfast, UK, DU-888U3-CSO-#BV), with laser illumination at 405 and 488 nm, and were analyzed using NIS-elements software (version 5.11.01, Nikon); 41 z planes per site spanning 20 μm were acquired. After acquisition, the images were filtered with a one-pixel-diameter median filter, subjected to background subtraction via a rolling ball radius of 50 pixels, and further subjected to maximum intensity projection using ImageJ software. The processed images were segmented into cell nuclei using Cellpose (version 2.1.0)[57], and the average intensity of nuclear mNG was calculated. These cells were further expanded, and proteins were extracted from the cells for western blot analysis.

### STREAMING-tag knock-in by genome editing

First, we describe the establishment of *Nanog* STREAMING-tag knock-in cells. C57BL/6 J mESCs ($5 \times 10^5$) were plated into each well of a 12-well plate; after 1 h, the following transfection reagents were mixed: 2 μg of targeting vector (e.g., pTV-Nanog_2-5prime-1000-24MS96T-3F_NeoR), 700 ng of CRISPR vector (e.g., eSpCas9-EF-5Nanog_2), and 300 ng of pKLV-PGKpuro2ABFP. Subsequently, 62.5 μL of Opti-MEM-reduced serum medium (Life Technologies, 11058021) and 2.5 μL of P3000 reagent (Life Technologies, L3000015) were added to each plate. In a separate tube, 62.5 μL of Opti-MEM-reduced serum medium and 4.5 μL of Lipofectamine 3000 (Life Technologies, L3000015) were added per reaction and mixed well. The P3000 and Lipofectamine 3000 media were mixed in equal volumes and incubated for 15 min at room

temperature (RT). The complex was then added to the wells containing the cells and incubated overnight. After 24 h, the medium was replaced with fresh 2i medium containing 1 µg/mL puromycin (Wako, 160-23151) to eliminate untransfected cells. After 24 h, the medium was replaced with fresh 2i medium; 24 h later, all cells were transferred to gelatin-coated 10 cm dishes. After 48 h, the medium was replaced with 2i medium containing 200 µg/mL G418; 48 h later, the medium was replaced with 2i medium containing 200 µg/mL G418. After another 48 h, 24 colonies were selected for further analysis. Genomic DNA was extracted from these cells, and genomic PCR was performed to narrow down the candidate cell lines. Thereafter, candidate clones were analyzed using Southern blotting, as described previously[32]. The restriction enzymes and genomic regions used for the Southern blot probes are summarized in Supplementary Data 3. Probes were prepared using the PCR DIG Probe Synthesis Kit (Roche, Basel, Switzerland, 11636090910).

The same procedure was used to knock-in the STREAMING-tag in *Pou5f1*, *Sox2*, *Wnk1*, *Flnc*, *Usp5*, and *Nanog* in PP7/ + cell line. In this cell line, the PP7 repeat monoallelically knocked-in to the *Nanog* 3′ UTR (TK-2A-Puro cassette was removed from TP/ + cells[32] by transient expression of Cre recombinase). The plasmids used are listed in Supplementary Data 1 and 2, respectively.

### Establishment of fluorescent protein-expressing cells

NSt-GR cells were established as follows: NSt mESCs ($2.5 \times 10^5$) were plated into each well of a 24-well plate, and after 1 h, the following transfection reagents were mixed. In a tube, 50 ng pCAG hyPBase and 75 ng mTetR-mNG expression vectors (such as pLR5-CAG-TetR_W43F-3xmNG) and 375 ng of pLR5-CAG-hMCP-mScarlet-I-NLS were mixed. Next, 25 µL of reduced serum Opti-MEM and 1 µL of P3000 reagent were added to each plate. In a separate tube, 25 µL of reduced serum Opti-MEM and 1.8 µL of Lipofectamine 3000 were added per reaction and mixed well. The P3000 and Lipofectamine 3000 media were mixed in equal volumes and incubated at RT for 15 min. The medium was replaced with fresh 2i medium after 24 h (day 2). Every 24 h, the medium was replaced with fresh 2i medium. On day 5, all cells were passaged in 12-well plates, and the cells were collected on day 6 following treatment with trypsin. Cells moderately expressing mTetR-mNG and MCP-RFP were isolated using a BD FACSAria III cell sorter (BD Biosciences, Franklin Lakes, NJ, USA), operated using BD FACSDiva Software (version 8.0.1), and plated into gelatin-coated 6 cm dishes (Supplementary Methods 1, Supplementary Fig. 10). At 8 days after fluorescence-activated cell sorting (FACS), colonies were picked and plated into a gelatin-coated 8-well chambered cover glass. Three days later, cells expressing moderate amounts of fluorescent protein, referred to as NSt-GR cells, were observed using fluorescence microscopy and used for further experiments.

For transfection with transgene containing SNAPtag, the following method was used: NSt mESC or other cell lines ($2.5 \times 10^5$) were plated into each well of a 24-well plate, and after 1 h, the following transfection reagents were mixed: 50 ng pCAG hyPBase, 75 ng pLR5-CAG-TetR_W43F-3xmNG, 275 ng pLR5-CAG-hMCP-mScarlet-I-NLS, and 100 ng of SNAPtag expression vector (e.g., pLR5-CAG-NLS-SNAP). To each of these, 25 µL of reduced serum Opti-MEM and 1 µL of P3000 reagent were added. In a separate tube, 25 µL of Opti-MEM reduced serum medium and 1.8 µL of Lipofectamine 3000 were added per reaction and mixed well. The P3000 and Lipofectamine 3000 media were mixed in equal volumes and incubated at RT for 15 min. After 24 h (day 2), the medium was replaced with 2i medium. After another 24 h (day 3), the medium was replaced with 2i medium, and the cells were passaged in 12-well plates on day 5. On day 6, the cells were incubated in 2i medium containing 300 nM SNAP-Cell 647-SiR (New England Biolabs, Ipswich, MA, USA, S9102S) for 30 min at 37 °C and 5% $CO_2$. The cells were washed three times with 2i medium and incubated at 37 °C and 5% $CO_2$ for another 30 min. The cells were collected following

treatment with trypsin, and cells moderately expressing mTetR-mNG, MCP-RFP, and SNAPtag were sorted using a BD FACSAria III cell sorter and seeded into gelatin-coated 6 cm dishes (Supplementary Methods 1, Supplementary Fig. 10). The medium was changed once every 2 days. At 8 days after FACS sorting, colonies were picked and plated into a gelatin-coated 8-well chambered cover glass and cultured. After 3 days, cells expressing a moderate amount of fluorescent protein were observed under a fluorescence microscope and used for further experiments. The cells were expected to show a mild level of fluorescence expression; if the expression level was too high, it was difficult to detect spots and foci (Supplementary Methods 1, Supplementary Fig. 10).

### RF-SNAPtag knock-in

NSt-GR mESCs ($1.25 \times 10^5$) were plated into each well of a 24-well plate, and after 1 h, the transfection reagents were mixed. In a tube, 500 ng of targeting vector (e.g., Rpb1 snap targeting vector), 250 ng of CRISPR vector (e.g., eSpCas9-Rpb1-gRNA), and 75 ng of pKLV-PGKpuro2ABFP were mixed. To each of these, 31 µL of reduced serum Opti-MEM and 1.25 µL of P3000 reagent were added. In a separate tube, 31 µL of reduced serum Opti-MEM and 2.25 µL of Lipofectamine 3000 were added per reaction and mixed well. The P3000 and Lipofectamine 3000 media were mixed in equal volumes and incubated at RT for 15 min. After 24 h (day 2), the cells were treated with 1 µg/mL puromycin in 2i medium. The medium was replaced with fresh 2i medium after another 24 h (day 3). Every 24 h, the medium was replaced with fresh 2i medium. On day 6, the cells were incubated for 30 min in 2i medium containing 300 nM SNAP-Cell 647-SiR at 37 °C and 5% $CO_2$. The cells were washed three times with 2i medium and incubated at 37 °C and 5% $CO_2$ for another 30 min. The cells were collected by trypsin treatment, and SNAPtag signal-positive cells were sorted using a BD FACSAria III cell sorter and seeded into gelatin-coated 6 cm dishes. The medium was changed once every 2 days. Twenty-four colonies were picked on day 8 after FACS. Genomic DNA extracted from these cells was used for genomic PCR to narrow down the candidate cell lines. Candidate clones were further analyzed using Southern blotting, as described previously[32]. The restriction enzymes and genomic regions used for the Southern blot probes are summarized in Supplementary Data 3. Probes were prepared using a PCR DIG Probe Synthesis Kit (Roche Diagnostics, Mannheim, Germany).

The SNAPtag was knocked into *Brd4*, *Med19*, and *Med22*, as described above. See Supplementary Data 1 and 2 for the plasmids used in this study. We confirmed that these SNAPtag knock-in cell lines have a growth rate comparable to that of the parental cell lines, suggesting that the effect of SNAPtag knock-in on the cells is negligible.

### Microscopy

For live cell imaging of mESCs, the medium was replaced with imaging 2i medium (FluoroBrite DMEM [Thermo Fisher Scientific, A1896701], 15% FBS [GE Healthcare, SH30396.03], 0.5 mM monothioglycerol solution [Wako, 195-15791], 1× MEM nonessential amino acids [Wako, 139-15651], 2 mM L-alanyl-L-glutamine solution [Wako, 016-21841], 1,000 U/mL LIF [Wako, 195-16053], 20 µg/mL gentamicin [Wako, 078-06061], 3 µM CHIR99021 [Cayman Chemical, 13122], 1 µM PD0325901 [Chemscene, CS-0062], and VectaCell Trolox Antifade Reagent for Live Cell Imaging [Vector Laboratories, Burlingame, CA, USA, CB-1000; 1:1,000]). For single-molecule fluorescence in situ hybridization (smFISH), the samples were mounted in catalase/glucose oxidase-containing mounting media (GLOX; 0.4% glucose [Nacalai Tesque, 16806-25] in 10 mM Tris-HCl [pH 8.0], 2× saline sodium citrate [SSC], glucose oxidase [37 µg/mL, Sigma-Aldrich, G2133-10KU], and 1/100 catalase [Sigma-Aldrich, C3155]). Images were acquired using a Nikon Ti-2 microscope with a CSU-W1 confocal unit, a 100× Nikon Apo TIRF oil-immersion objective lens (NA 1.49), and an iXon Ultra EMCCD (Andor Technology), operated using NIS-Elements software (ver.

5.11.01; Nikon). The microscope was also equipped with 405, 488, 561, and 637 nm lasers (Andor Technology), a stage-top microscope incubator for live cells (5% $CO_2$; 37 °C; STXG-TIZWX-SET, Tokai Hit, Shizuoka, Japan), and an ASI MS-2000 piezo stage (ASI). Z-stack images spanning 20 μm with 200 nm intervals (101 sections; 130 nm/pixel) were acquired.

## Snapshot fluorescence imaging of live cells

Cells ($5 \times 10^4$) were plated onto each well of an 8-well chambered cover glass (Cellvis) that was pre-coated with laminin-511 (BioLamina, Sundbyberg, Sweden, BLA-LN511-0502) and cultured overnight. For imaging SNAPtag and HaloTag, the cells were incubated in 2i medium containing 300 nM SNAP-Cell 647-SiR and Janelia Flour 646 HaloTag ligand, respectively, for 30 min at 37 °C and 5% $CO_2$, washed three times with fresh 2i medium, and incubated for another 30 min at 37 °C and 5% $CO_2$. The medium was then replaced with imaging 2i medium (FluoroBrite DMEM [Thermo Fisher Scientific, A1896701] containing VectaCell Trolox Antifade Reagent for Live Cell Imaging (Vector Laboratories, CB-1000; 1:1,000). After image acquisition, the images were processed with a one-pixel diameter 3D median filter using ImageJ software (NIH, Bethesda, MD, USA).

## Western blotting

Cells were washed twice with phosphate-buffered saline (PBS, Nacalai Tesque, 14249-24), trypsinized, and collected by centrifugation at 190 × g for 2 min at 20 °C. The cells were counted and washed twice with PBS. The cells were then lysed in lysis buffer (0.5% Triton X-100 (Sigma-Aldrich, T8787-100ML), 150 mM NaCl (Wako, 191-01665), and 20 mM Tris-HCl [pH 7.5]) to obtain $2 \times 10^6$ cells per 100 μL. The lysates were then incubated at 95 °C for 5 min and filtered using a QIAshredder homogenizer (Qiagen, 79656). The extracted proteins were analyzed using 5–20% gradient sodium dodecyl sulfate-polyacrylamide gel (SDS-PAGE) electrophoresis and transferred onto Immobilon Transfer Membranes (Millipore, INYC00010) for immunoblotting. The primary antibodies used were anti-mNeonGreen (1:500; Chrom Tech, Apple Valley, MN, USA, 32f6-100, RRID:AB_2827566), anti-GAPDH (1:5000; Cell Signaling Technology, Danvers, MA, USA, 5174, RRID:AB_10622025), anti-NANOG (1:1000; eBioscience, San Diego, CA, USA, 14-5761-80, RRID:AB_763613), anti-SOX2 (1:1000; Abcam, Cambridge, UK, ab97959, RRID:AB_2341193), and anti-USP5 (1:2000; 10473-1-AP, Proteintech, Rosemont, IL, USA, RRID:AB_2272754).

## smFISH

Cells ($5 \times 10^4$) were transferred onto a laminin 511-coated 8-well chambered cover glass and cultured for 1 h at 37 °C and 5% $CO_2$. The cells were washed with PBS, fixed with 4% paraformaldehyde (Wako, 168-20955) in PBS for 10 min, and washed twice with PBS. The cells were then permeabilized in 70% ethanol (Wako, 054-07225) at 4 °C overnight. After washing with 10% formamide (Wako, 066-02301) dissolved in 2× SSC (Nacalai Tesque, 32146-04) buffer, the cells were hybridized to probe sets in 130 μL of hybridization buffer containing 2× SSC, 10% dextran sulfate (Nacalai Tesque, 03879-72), 10% formamide, and 1 μM of primary probes (Supplementary Data 4). Hybridization was performed for 12 h at 37 °C in a moist chamber. The coverslips were washed with 10% formamide in 2× SSC solution and incubated at 37 °C for 30 min in the dark. The cells were hybridized to probe sets in 130 μL of hybridization buffer containing 2× SSC, 10% dextran sulfate, 10% formamide, and 125 nM secondary probe (Supplementary Data 4) with or without and *Nanog* Exonic probes[32]. Hybridization was performed for 4 h at 37 °C in a moist chamber. The coverslips were washed with 10% formamide in 2× SSC solution, incubated at 37 °C for 30 min in the dark, and then washed with 10% formamide in 2× SSC solution with Hoechst 33342 (1:1000, Thermo Fisher Scientific, H3570) and incubated at 37 °C for 30 min in the dark. Hybridized cells were mounted in GLOX buffer. After image acquisition, the images were filtered with a

one-pixel-diameter three-dimensional median filter and subjected to background subtraction via a rolling ball radius of 5 pixels using ImageJ software. Detection and counting of smFISH signals and estimation of the number of nascent RNA in the transcription sites were performed using Big-FISH (version 0.6.2)[58].

## FRAP analysis

NSt-MT(WT) and NSt-MT(W43F) cells (Supplementary Data 1) ($5 \times 10^4$) were plated into each well of laminin-511-coated 8-well chambered cover glass and cultured overnight. The medium was replaced with an imaging medium (2i). For photobleaching, a fluorescence recovery after photobleaching (FRAP) module (Nikon) was used in combination with a CSU-W1 confocal system. Five z-stack images (17 sections at 0.2 μm intervals) were taken at 4 s intervals, and after applying a 488 nm laser pulse (100 ms; 24.2% laser attenuation; 50 mW laser output) through a FRAP module, 25 z-stack images (17 sections at 0.2 μm intervals) were taken at 4 s intervals. The images were processed with a one-pixel-diameter 3D mean filter and subjected to maximum intensity projection and "bleach correction" using ImageJ software. After manually selecting the center of the TetR regions, the average fluorescence intensities of a circular region of interest of 6 pixels in diameter with the center were measured.

The normalized Intensity $I_{norm}$ was calculated using the following equation:

$$I_{norm} = I/I_{pre} \tag{1}$$

where, $I_{pre}$ is the fluorescence intensity of pre-FRAP. To extract the characteristic timescales of fluorescence recovery and the mobile fractions, average FRAP curves were fitted using R with the following function:

$$I_{norm}(t) = baseline + f_{mobile}(1 - e^{-t/\tau}) \tag{2}$$

where, *baseline*, $\tau$, and $f_{mobile}$ represent the expected $I_{norm}(0)$, recovery time constant, and mobile fraction, respectively. Assuming that the *baseline* was the same for TetR(WT) and TetR(W43F)/mTetR, the sum of the squares of the offsets ($S$) was used to estimate the best-fit curve. $S_{WT} + S_{W43T}$ was minimized when the *baseline* value was 0.41.

## Effect of TetR on transcription of STREAMING-tag

To confirm the effect of TetR on STREAMING-tag transcription, we evaluated the effect on transcription using cell lines derived from NSt cells (Supplementary Data 1) and expressing MCP-RFP and either TetR(W43F)-mNG, TetR(WT)-mNG, or NLS-mNG. These cell lines were plated in an 8-well chambered cover glass (Cellvis) coated with laminin 511 ($8 \times 10^4$ cells/well). The next day, cells were imaged as describe in the "Snapshot fluorescence imaging of live cells" subsection. After image acquisition, the images were filtered with a one-pixel-diameter gaussian filter and subjected to background subtraction via a rolling ball radius of 5 pixels using ImageJ software. The processed images were segmented into cell nuclei using Cellpose[57], and the average intensity of MCP-RFP nuclei was calculated. A single candidate MCP-RFP spot per cell was identified by the local maxima using trackpy.locate. The RFP relative intensity values of MCP-RFP spots to the RFP average intensity value in the cell nucleus were calculated. In the histogram of the relative intensity values of MCP-RFP spots, the valley between the distribution of small values and larger values was located at approximately 3, so that transcription was assumed to be ON when the value was 3 or higher.

## ChIP-qPCR

ChIP-qPCR was performed using the SimpleChIP® Enzymatic Chromatin IP Kit (Cell Signaling Technology, #CST 9003 S) using the manufacturer's protocol. For this experiment, cell lines derived from

NSt cells (Supplementary Data 1) and expressing MCP-RFP and either TetR(W43F)-mNG, TetR(WT)-mNG, or NLS-mNG were used. Approximately $1 \times 10^7$ cells were used per experiment, with 20% of each sample as input after MNase digestion. For immunoprecipitation, 2 μg of mouse anti-RPB1 unphosphorylated CTD antibody (CMA601, in house, 1:200) was used[59]. DNA purified from immunoprecipitated chromatin was subjected to qPCR analysis using THUNDERBIRD Next SYBR qPCR (TOYOBO). Primer sets containing a positive control (*Gapdh* promoter) was used in the qPCR analysis (Supplementary Table 1). The Ct value of each sample was used to calculate % input from 2^[(Ct input −5.64) -Ct sample]×100. In addition, fold enrichment (% to *Gapdh*) was calculated by normalizing the % input values between samples by the *Gapdh* values.

## DNA-FISH

The NSt-GR cells (Supplementary Data 1) were plated onto laminin-511-coated glass slides and cultured overnight at 37 °C and 5% $CO_2$. The cells were washed with PBS, fixed with fresh 4% paraformaldehyde in PBS for 10 min, washed twice with PBS, and then treated with PBS containing Hoechst 33342 nucleic acid stain (1:1000) for 10 min. Images were acquired using a Nikon Ti-2 microscope with a CSU-W1 confocal unit, a 100× Nikon Apo TIRF oil-immersion objective lens (NA 1.49), and an iXon Ultra EMCCD camera with laser illumination at 405, 488, and 637 nm. In this setup, the pixel size was 130 nm, and 76 z-planes per site spanning 15 μm (z-step = 200 nm) were acquired. The cells were then subjected to 3D-DNA-FISH as previously described with some modifications[60]. Briefly, the cells were washed twice with PBS and then permeabilized in 0.1% saponin (Nacalai Tesque, 30502-42)/0.1% Triton X-100/2 mM ribonucleoside vanadyl complex (New England Biolabs, S1402S) in PBS for 10 min at RT. Following two washes with PBS, the cells were incubated for 20 min in 20% glycerol/PBS at RT and stored in 50% glycerol/PBS at −20 °C for at least 1 day. After incubation, the cells were recalibrated at RT in 20% glycerol/PBS and subjected to three successive freeze/thaw cycles in liquid nitrogen[60]. Thereafter, the cells were washed twice with PBS for 5 min each at RT, incubated in 0.1 M HCl (Nacalai Tesque, 18320-15) for 30 min at RT, washed once again with PBS for 5 min at RT, permeabilized in 0.5% saponin/0.5% Triton X-100 in PBS for 30 min at RT, washed two more times with PBS for 5 min per wash at RT, and then equilibrated in 50% formamide/2× SSC for 10 min at RT. Next, the cells were hybridized to a predenatured *Nanog* probe (see below) using a hybridization buffer containing 1× SSC, 10% dextran sulfate, and 50% formamide. Hybridization was performed for 16 h at 37 °C in a moist chamber. The cells were washed in 2× SSC for 5 min at RT, 50% formamide/2× SSC for 15 min at 45 °C, 2× SSC for 5 min at 45 °C, and then 2× SSC for 5 min at RT, followed by a wash in 2× SSC with Hoechst 33342 (1:1000) for 10 min at RT. Hybridized cells were mounted in GLOX buffer[32]. Images were acquired using a laser illumination set at 405 nm for Hoechst 33342 and at 647 nm for *Nanog* probes. The BAC clones RP23-19O18 (CHORI BACPAC Resources, Emeryville, CA, USA) were used as DNA-FISH probes for *Nanog*.

After image acquisition, the images were processed with a one-pixel diameter 3D median filter and registered with "Correct 3D drift" based on the Hoechst channel using Fiji. DNA-FISH images were corrected for cell nucleus deformation based on images obtained immediately after cell fixation using Fijiyama[61]. DNA-FISH and mTetR spots as well as cell nuclei were detected from maximum intensity projected images using Imaris software (version 9.1.2, Bitplane, Zürich, Switzerland). The distance between spots in the same cell was calculated.

## Analysis of MS2 RNA recognition specificity of MCP

The specificity of MS2 RNA recognition by MCP was verified using MCP mutants. S47R and R49H mutations reportedly significantly inhibit MS2 binding of MCP[38,39]. Therefore, we established cells derived from NSt cells that stably express mTetR-mNG and either MCP-RFP,

MCP(S47R)-RFP, or MCP(S47R, R49H)-RFP (Supplementary Data 1). We also established cells derived from WT cells that stably express mTetR-mNG and MCP-RFP (Supplementary Data 1). To quantify RFP fluorescence intensity at mTetR spots, $8 \times 10^4$ cells were plated on laminin-coated 8-well chambered cover glass with #1.5 glass (Cellvis). The cells were imaged the next day as described in the "Snapshot fluorescence imaging of live cells" subsection. After acquisition, images were filtered with a one-pixel-diameter gaussian filter, subjected to background subtraction via a rolling ball radius of 5 pixels, and subjected to maximum intensity projection using ImageJ software. The processed images were segmented into cell nuclei using Cellpose[57], and the average intensity of MCP-RFP nuclei was calculated. A single candidate mTetR spot per cell was identified by the local maxima using trackpy.locate. The RFP relative intensity values of MCP-RFP spots to the RFP average intensity in the cell nucleus were calculated. The following analysis was performed to analyze the specificity of MCP binding sequences to MS2 RNA. The process from cell plating to image processing was described in the "Analysis of MS2 RNA recognition specificity of MCP" subsection. Candidate MCP spots were identified by the local maxima using trackpy.locate (diameter = 5, topn = 2000). The RFP relative intensity values of MCP-RFP spots to the RFP average intensity in the cell nucleus were calculated. The number of MCP-RFP spots per cell with a relative intensity value of 2, 3, 4, 5, or higher was calculated.

## SNR analysis

The NMP-R mESCs[27] ($5 \times 10^4$) (Supplementary Data 1) were plated into 24-well plates on the day before transfection. The cells were transfected with 700 ng of MS2 sgRNA expression vectors[27] on the following day using Lipofectamine 2000 (Thermo Fisher Scientific, 11668019). After 12 h, the cells were treated with puromycin (2 μg/mL) and doxycycline (100 ng/mL, MP Biomedicals, Santa Ana, CA, USA, 195044) for another 24 h. The cells were trypsinized, transferred onto a laminin-511-coated 8-well chambered cover glass (CellVis, C8-1.5H-N), and cultured overnight at 37 °C and 5% $CO_2$ in doxycycline-containing medium.

For NSt-GR cells, an 8-well chambered cover glass (CellVis, C8-1.5H-N) coated with laminin-511 was used. Cells ($5 \times 10^4$) were plated into each well and cultured at 37 °C and 5% $CO_2$ overnight.

After image acquisition, the area outside the cell nucleus was measured as the average background intensity, and the intensity was subtracted from the entire image using ImageJ software. The standard deviation ($\sigma_N$) was measured in the nucleus. Next, the DNA-labeled region was selected using the "Find Maxima" function of TrackMate with an estimated blob diameter of 0.5 μm, and the mean intensity value (μ) of the target foci was measured. The signal-to-noise ratio (SNR) was calculated using the following equation:

$$SNR = \mu/\sigma_N \qquad (3)$$

## MSCD and MSD analysis

For MSCD analysis, NSt-NLS-SNAP, PSt-NLS-SNAP, SSt-NLS-SNAP, USt-NLS-SNAP, WSt-NLS-SNAP, and FSt-NLS-SNAP mESCs ($5 \times 10^4$) (Supplementary Data 1) were plated into an 8-well chambered cover glass with laminin-511 and cultured overnight at 37 °C in an atmosphere of 5% $CO_2$. The cells were incubated in 2i medium containing 300 nM SNAP-Cell 647-SiR for 30 min at 37 °C and 5% $CO_2$. The cells were then washed three times with 2i medium and incubated for another 30 min at 37 °C and 5% $CO_2$. After the medium was replaced with imaging 2i medium, 46 z-sections spanning 9 μm (z-step = 200 nm) were acquired at 15 s intervals for 450 s. The acquired images were filtered with a one-pixel diameter 3D Gaussian Blur filter using ImageJ software. The mTetR-mNG images were subjected to background subtraction using a rolling ball radius of five pixels using ImageJ software. Fluorescent spots were detected using "Spot" function of Imaris software (version

9.1.2, Bitplane, Zürich, Switzerland) with the spot diameter set to 0.8 μm (semi-automatic detection). The nucleus center of mass was determined from NLS-SNAP fluorescence using ImarisCell (Bitplane), with the Cell Smooth Filter Width and Cell Background Subtraction Width parameters set at 1 and 0.64 μm, respectively. The mean square change in distance (MSCD) was calculated as the average change in distance between the nuclear center of mass and genomic locus over all possible combinations of time points separated by the lag time $\Delta t$; $MSCD = [d(t) - d(t + \Delta t)]^2$[2,41,42]. To categorize the "ON" and "OFF" states, the threshold was set based on the histogram of fluorescence intensities in the mTetR-mNG area in the MCP channel of all analyzed images. A valley was typically found between a peak at the background level and broad peaks at higher intensities, and the value at the valley (*fluorescence intensity* = 5) was used as the threshold. If transcription was observed at least once during the time-lapse, the data were classified as "ON."

For MSD analysis, NSt-GR and SSt-GR mESCs ($5 \times 10^4$) were plated into an 8-well chambered cover glass with laminin-511 and cultured overnight at 37 °C in an atmosphere of 5% $CO_2$. After the medium was replaced with imaging 2i medium, MCP-RFP images were obtained first. Then, 400 frames of mTetR-mNG images with 30 ms exposure time were acquired without intervals. The acquired images were filtered with a one-pixel diameter Gaussian Blur filter and subjected to background subtraction using a rolling ball radius of five pixels using ImageJ software. First, a rectangle region of interest (ROI) centered on the mTetR spot was selected manually. A single candidate mTetR spot per ROI were determined by local maxima using Trackpy (version 0.5.0). The detected spots were linked together using trackpy link (size = 2, memory = 4). Only ROIs in which mTetR spots were detected in more than 50 consecutive frames were used for later analysis. Time- and ensemble MSD was calculated using Trackpy. The MSD up to 66 steps (2 s) was fitted to an anomalous diffusion model of the form: $MSD(t) = 4D_\alpha t^\alpha$, where $D_\alpha$ is the apparent diffusion constant, and $\alpha$ is the anomalous coefficient ($0 < \alpha < 2$). The RFP relative intensity values at mTetR-mNG spots to the RFP average intensity value in the cell nucleus were calculated. In the histogram of the relative intensity values of MCP-RFP spots, the valley between the distribution of small values and larger values was located at approximately 3, so that transcription was assumed to be ON when the value was 3 or higher.

### Transcriptional kinetic analysis

NStP-GRH mESCs ($8 \times 10^4$) (Supplementary Data 1) were plated into an 8-well chambered cover glass with laminin-511 and cultured overnight at 37 °C in an atmosphere of 5% $CO_2$. The cells were incubated in 2i medium containing 300 nM Janelia Flour 646 HaloTag ligand for 30 min at 37 °C in an atmosphere of 5% $CO_2$. The cells were then washed three times with 2i medium and incubated for another 30 min at 37 °C in an atmosphere of 5% $CO_2$. After the medium was replaced with imaging 2i medium, 25 z-sections spanning 10 μm (z-step = 400 nm) of MCP-RFP and PCP-HaloTag channels were acquired at 1 min intervals for 4 h. The acquired images were filtered with a one-pixel diameter Gaussian Blur filter, subjected to background subtraction using a rolling ball radius of 50 pixels, subjected to maximum intensity projection, and with "Correct 3D drift" based on the PCP-HaloTag channel using Fiji software. Cell nuclei in the processed images were segmented using Cellpose (version 2.1.0)[57], and the center coordinates of cell nuclei were associated between frames. MCP and PCP spots were detected using trackpy.locate. The relative fluorescent intensities of MCP and PCP spots to the average fluorescent intensity of cell nuclei were calculated. Cross-correlation functions were computed and averaged as described previously[44].

### Cloning antibody variable fragments encoding 44B12

Mouse hybridoma cells expressing Ser5ph-specific antibodies were generated (MAB Institute, Inc., Nagano, Japan) as previously

described[62], using a peptide Ser-Tyr-Ser-Pro-Thr-phosphoSer-Pro-phosphoSer-Tyr-Ser-Pro-Thr-Ser-Pro-Ser-Tyr-Ser-Pro-Cys, harboring Ser5ph and Ser7ph at the C-terminal domain sequence. The resulting antibody 44B12 reacted with peptides containing Ser5ph, regardless of the phosphorylation state of Ser7 (Supplementary Fig. 9). To construct the mintbody, RNA was purified using TRIzol (Thermo Fisher Scientific, 15596026), and the sequences encoding IgG heavy and light chains were determined using RNA sequencing[63]. The variable regions of heavy and light chains ($V_H$ and $V_L$) were each amplified using PCR with specific primers (for VH, 44B12-VH_s (5′-CGAATTCGCCATGGCCGA GGTCCAGCTGCAACAGTC-3′) and 44B12-VH_as (5′-TGAACCGCCTCCA CCTGCAGAGACAGTGACCAGAG-3′); for VL, 44B12-VL_s (5′-TCTGGC GGTGGCGGATCGGATGTTGTGATGACCCAGAC-3′) and 44B12-VL_as (5′-TGGATCCGCCCGTTTGATTTCCAGCTTG-3′)) and then connected using linker primers (LINK primer1 (5′-GTCTCCTCAGGTG GAGGCG GTTCAGGCGGAGGTGGCTCTGGCGGTGGCGGATCG-3′), LINK primer2 (5′-CGATCCGCCACCGCCAGAGCCACCTCCGCCTGAACC GCCTCCAC CTGAGGAGAC-3′)) as described previously[46,64]. The scFv fragment was cloned into the sfGFP-N1 vector (Addgene #54737)[65] using In-fusion (Takara, Shiga, Japan) with the primers 44B12-scFv_s (5′-CTCGAGCTC AAGCTTCGAATTCGCCATGGCCGAAG-3′) and 44B12-scFv_as (5′- CAT GGTGGCGACCGGTGGATCCGCCCGTTTTATTTCCAG-3′) to generate a 44B12-sfGFP expression vector.

Two-point mutations predicted to improve the folding and/or stability of scFv were introduced into the RNAPII Ser5ph-mintbody as described previously[47]. Mintbodies were expressed in HeLa cells by transfecting the expression vectors using Fugene HD (Promega, Madison, WI, USA, E2311). A plasmid vector (2 μg) and Fugene HD (6 μL) were mixed in Opti-MEM (Thermo Fisher Scientific; 100 μL). After incubation at RT for 30 min, the mixture was added to HeLa cells grown in 35 mm glass-bottom dishes (AGC Technology Solutions, Kawasaki, Japan). The culture medium was changed to FluoroBrite DMEM containing 10% FBS and 1% GPS, and fluorescence images were acquired using a point-scan confocal microscope (Olympus, Tokyo, Japan, FV1000 with IX-81) with a UPlanApoN 60× OSC oil immersion objective lens (NA 1.4) using the built-in software FLUOVIEW ver. 4.2 (512 × 512 pixels, pixel dwell time 4.0 μs, pinhole 100 μm, zoom ×5.0, line averaging ×4, and a multi-argon ion 488 nm laser line with 10% transmission).

### Purification of RNAPII Ser5ph-mintbody

RNAPII Ser5ph-mintbody was purified using the same method as used for the Ser2ph-mintbody[47]. Briefly, a His-tag-RNAPII Ser5ph-mintbody expression vector was constructed using pTrc-His (Thermo Fisher Scientific, V36020). *Escherichia coli* BL21 (DE3) cells harboring the expression vector were grown in YTG medium (1% tryptone, 0.5% yeast extract, 0.5% NaCl, and 0.2% glucose; 100 mL) for 20 h at 18 °C. After dilution in YTG medium (2 L), the cells were incubated for 8 h at 15 °C, followed by incubation with isopropyl-β-D-thiogalactopyranoside (at a final concentration of 1 mM) for 12 h at 15 °C. The cells were collected by centrifugation (4,000 ×g; 10 min; 4 °C), and the pellet was stored at −80 °C until thawing in 20 mL Buffer L (50 mM Tris-HCl, pH 8.0, 300 mM NaCl, 5% glycerol) containing 1 mg/mL lysozyme (Nacalai Tesque) and 1% proteinase inhibitor cocktail (Nacalai Tesque). After sonication (Branson Ultrasonics, Brookfield, CT, USA; Sonifier 250) for lysis, cell debris was removed by centrifugation (15,000 ×g for 15 min at 4 °C). Ni-NTA agarose (Qiagen, Hilden, Germany; 0.5 mL) was equilibrated with Buffer L and settled in an open column (Poly-Prep Chromatography Columns; Bio-Rad, Hercules, CA, USA; 20 mL) before the cell lysate was applied at 4 °C. After washing the column twice with Buffer L (10 mL each), elution buffer (Buffer L containing 150 mM imidazole, pH 8.0) was added (1 mL, three times). The eluted fractions were dialyzed against starting buffer (10 mM Tris-HCl, pH 7.0, 50 mM NaCl; 1 L) overnight with buffer exchange. The His-RNAPII Ser5ph mintbody was further purified using a HiTrap Q column (GE Healthcare) with a linear gradient elution with End Buffer (10 mM Tris-HCl,

1 M NaCl) using AKTAprime plus (GE Healthcare) at 4 °C. After 10–20% SDS-PAGE and Coomassie Blue staining, fractions containing His-RNAPII Ser5ph-mintbody were pooled. After the His-tag was removed using an enterokinase cleavage capture kit (Novagen, Madison, WI, USA), His-tag- and enterokinase-free RNAPII Ser5ph-mintbody was prepared by passing through EKapture™ Agarose Millipore).

### ELISA

Enzyme-linked immunosorbent assay was performed as previously described[47]. Microtiter plates (Greiner Bio-One, Kremsmünster, Austria, 655061) were coated with 1 μg/mL bovine serum albumin conjugated with RNAPII C-terminal domain peptides with or without phosphorylated amino acids (MAB Institute, Inc.; Supplementary Fig. 9) overnight at 4 °C. The plates were washed three times with PBS (Fujifilm Wako Pure Chemical, Osaka, Japan, 048-29805) containing 0.1% Tween-20 (Fujifilm Wako Pure Chemical, 167-11515) (PBST), and each well was incubated with Blocking One P (Nacalai Tesque, 05999-84; 100 μL) for 20 min at RT, washed three times with PBST, and incubated with a 1:3 dilution series of purified RNAPII Ser5ph-mintbody (starting at 300 ng/mL) and IgG antibodies (starting at 30 ng/mL) specific for Ser2ph (CMA602, RRID: AB_2819246)[59] and Ser5ph (CMA603, RRID: AB_2827955)[59] in PBST (100 μL) overnight at 4 °C. After washing three times with PBST, the plates were incubated with anti-GFP (1:2,000, MBL, 598-7) or anti-mouse IgG (1:10,000, Jackson ImmunoResearch, West Grove, PA, USA, 715-035-150), each conjugated with horseradish peroxidase for 120 min at RT. After washing three times with PBST and incubating for 10 min at RT, the plates were incubated in o-phenylenediamine solution (100 μL; 0.26 mg/mL in 0.1 M sodium citrate, pH 5.0, and 0.01% hydrogen peroxide; Fujifilm Wako Pure Chemical, 158-01671) at RT. The absorbance at 490 nm was measured with a reference wavelength of 600 nm using a Varioskan spectrophotometer (Thermo Fisher Scientific).

### Transcription inhibitor treatment

Nst-SNAP-Ser5ph cells (Supplementary Data 1) stably expressing RNAPII Ser5ph mintbody-SNAPtag were established as described above (for details see Establishment of fluorescent protein-expressing cells using the piggyBac system). Nst-SNAP-Ser5ph cells grew normally, suggesting that expression of the RNAPII Ser5ph mintbody did not significantly affect cell growth. Nst-SNAP-Ser5ph cells ($7 \times 10^4$) were cultured overnight on a laminin-511-coated 8-well chambered cover glass. Nst-SNAP-Ser5ph cells were incubated in 2i medium containing 300 nM SNAP-Cell 647-SiR for 30 min at 37 °C and 5% $CO_2$ to stain the SNAPtag. After incubation, the cells were washed three times with fresh 2i medium and further incubated for 30 min at 37 °C and 5% $CO_2$. The medium was replaced with imaging 2i medium containing 15 μM THZ1 (Selleck Chemicals, Houston, TX, USA, S7549), 1 μM flavopiridol (Chemscene LLC, CS-0018), 100 μM DRB, or 0.13% dimethyl sulfoxide, and the cells were further incubated for 1 h before imaging.

Acquired images were filtered with a one-pixel diameter 3D Gaussian Blur filter and subjected to background subtraction with a rolling ball radius of 50 pixels, followed by maximum intensity projection using ImageJ software. Nuclei were selected manually using "Polygon selections" or "Freehand selections" tools with the ROI manager, and the mean intensity and foci number of regions of interest were calculated using the find maxima (*prominence* = 10) and measure functions. The mean intensity of foci in the nucleus was determined by averaging the focus intensities of each cell.

### RF and mintbody imaging and analysis

SNAPtag knock-in cells or mintbody-SNAPtag-expressing mESCs ($5 \times 10^4$) were plated into a well of a laminin-511-coated 8-well chambered cover glass and cultured overnight at 37 °C and 5% $CO_2$. The cells were incubated in 2i medium containing 300 nM SNAP-Cell 647-SiR for 30 min at 37 °C and 5% $CO_2$. After washing the cells three times with 2i

medium, they were incubated for another 30 min at 37 °C and 5% $CO_2$. The medium was replaced with imaging medium (2i). The cells on the Laminin-511-coated 8-well chambered cover glasses were then imaged using microscope (see "live imaging setup" under "Microscopy" section for details). We focused on the z-position where the mTetR spot was detectable and acquired single-section images in the order of the SNAPtag, MCP, and mTetR channels ten times (exposure time for each channel was 40 ms; laser intensities are 43.9%, 23.8%, and 39.3% for 640 nm, 561 nm, and 488 nm lasers, respectively). To measure the accuracy of the microscope system, images of 0.1 μm fluorescent beads (TetraSpeck Microspheres, Thermo Fisher Scientific, T7279) on an 8-well chamber cover glass in imaging 2i medium were acquired. The images were processed using Gaussian Blur 3D (pixel size = 1) and bleach correction (simple ratio) using ImageJ software. Only SNAPtag images were processed with subtraction background rolling = 50, and all other images were processed with subtraction background rolling = 5. All images were then processed using average intensity projection.

Image analysis of SNAPtag cluster, MCP, and mTetR spots was performed as described by Li et al., 2020[8]. First, a $19 \times 19$-pixel region of interest centered on the mTetR spot was selected. The coordinates of mTetR and MCP spots were determined by local maxima using trackpy.locate in the Trackpy package (version 0.5.0). The spots were then subjected to Gauss fitting using trackpy.refine_leastsq to determine the coordinates at the subpixel level. Most signals in a single spot were covered in 6-pixel diameter circle (see Figs. 6b and 7a). Therefore, in this study, the mean intensities within a circle with a radius of 6 pixels from the spot center were calculated using Trackpy and were used as spot intensity values. MCP spots within 390 nm of the mTetR spot with a mean fluorescence intensity greater than two-fold to the mean intensity of the $19 \times 19$-pixel region of interest were classified as the "ON" state, and the remaining spots were classified as "OFF". SNAPtag clusters were identified by local maxima using trackpy.locate. In addition, the cluster centers were determined at the sub-pixel level by Gauss fitting using trackpy.refine_leastsq. The 2D distances between the nearest SNAP and mTetR coordinates were calculated in each cell.

To produce a RF cluster averaged image, $19 \times 19$ pixel images centered on the RF cluster coordinate that was closest to the mTetR spot were extracted and averaged. To measure the diameter (full width at half maximum peak height: FWHM) of RF clusters, the averaged images were processed using ImagJ plug-in GaussFit OnSpot (https://imagej.nih.gov/ij/plugins/gauss-fit-spot/index.html) with shape=-Circle, fitmode = [Levenberg Marquard] parameter.

### Statistics and reproducibility

The exact number, *n*, of data points and their representation (such as cells and independent experiments), and statistical tests used are indicated in the respective figure legends and in the results. All experiments were performed as two or more independent experiments. The same conclusions were obtained from each experiment. Statistical tests were performed in R software (The R Project for Statistical Computing, Vienna, Austria) and Python (Python Software Foundation, https://www.python.org/). Boxplots with descriptive statistics were created in R software and Python. Boxes indicate the interquartile range (IQR; 25–75% intervals) and median line; whiskers indicate 1.5-fold of the IQR.

### Reporting summary

Further information on research design is available in the Nature Portfolio Reporting Summary linked to this article.

## Data availability

Data supporting the findings of this study are available within the article, Supplementary Information and Source Data, and are also available from the corresponding author upon request. Source data are provided with this paper.

## Code availability

Python code used to measure the distance between the STREAMING-tag region and the RF cluster is available from GitHub under https://github.com/Ochiai-Lab/STREAMING-tag or https://doi.org/10.5281/zenodo.7328602

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

## Acknowledgements

We thank Ms. Yuki Ochiai for providing technical assistance and the Biomaterials Analysis Division, Open Facility Center, Tokyo Institute of Technology for DNA-sequencing analysis. We also thank L. Lavis (HHMI-Janelia) for dye-labeling reagents. This work was supported by Grants-in-Aid for Scientific Research from the Ministry of Education, Culture, Sports, Science, and Technology (JP18H05531, JP21H05753, JP22H02609, and JP22H04694 to H.Oc., JP22K15084 to H.Oh., JP21H04764 to H.K., JP20K06484 to Y.S, and JP18H05527 to Y.O. and H.K.) and the JST CREST program (JPMJCR16G1 to Y.O., H.K., and H.Oc.; and JPMJCR20S6 to Y.S).

## Author contributions

H.Oc. conceived, designed, and supervised the study and established all the knock-in cells. H.Oc., H.Oh. and S.S. performed verification experiments on the basic design of the STREAMING-tag. H.Oc., H.Oh., S.S., M.S., and H.Ow. established mintbody-SNAPtag and NLS-SNAPtag expressing cells and performed the MSCD experiments. H.Oh. performed STREAMING-tag and SNAPtag imaging experiments and analyzed the data. J.L. and A.P. developed the *Rpb1-*, *Brd4-*, *Med19-*, and *Med22*-SNAPtag knock-in vectors and their corresponding CRISPR vectors. S.U., Y.S., Y.O., and H.K. developed the RNAPII Ser5ph mintbody and performed validation experiments. H.K. supervised the development of mintbodies. H.Oc, H.Oh, T.Y., A.P., and H.K. interpreted the data and wrote the manuscript.

## Competing interests

The authors declare no competing interests.
