## [Peer Review File · Nature Communications]

REVIEWER COMMENTS

Reviewer #1 (Remarks to the Author):

The manuscript of Ohishi et al. describes a novel system named STREAMING-tag that allows for labeling of DNA and RNA at a single locus in a single integration step. The system is an upgrade of their previous ROLEX system, allowing the study of a gene locus both when it is OFF (transcriptionally inactive) or ON (transcriptionally active). Using this system, the authors show that clusters of RNA Polymerase, Brd4 and Mediator close to the locus. Conceptually, such a system could be highly valuable for the community and has the potential to study spatiotemporal aspects of transcription. However, there are a number of important concerns that limit its usability in its current form. In addition, the manuscript frequently lacks proper quantification and could benefit from more thorough analysis.

My major concern is that the tag appears to interfere with transcription, the process being studied:

1. The authors show a clear reduction in expression levels at all integrated genes. Tagging always has the risk of interfering with function. If there is a major loss of function from tagging, as is the case here, then one should hesitate to use the system. Instead, the author just state the loss of expression, and do not appear too concerned. Although the authors claim that this effect mostly comes from post-transcriptional effects, I am currently not convinced by the provided evidence. In Figure 2, the number of Nanog transcripts is shown, as well as the fraction of cells showing a transcription site in the tagged and untagged allele. However, the untagged allele is also mutated and could be reduced as well, as there is no fully functional allele in the cell. For Sox2, and perhaps for other genes, there is cellular feedback, where loss of function of one allele can affect both alleles. It is therefore important to also compare transcription of these cells to untagged wildtype cell lines? In addition, only burst frequency is analysed, but does the addition of the tag affect burst size (intensity of transcription site in smFISH)?

2. Moreover, the analysis in Fig 2 are performed in the absence of the TetR, and the authors show that TetR expression can majorly affect transcription in Figure 3. Although the authors optimize the TetR protein to reduce its affinity, they do not go on to show that transcription levels in their optimized system are similar with and without TetR(W43F) expression. The authors should directly compare the bursting kinetics (frequency and size) in cells without and without expression of TetR(W34F). In addition, they should check whether TetR(W43F) expression results in PolII stalling or premature termination, for example by Pol II-ChIP-seq.

3. In Figure 5 the authors show transcriptional dynamics quantification using STREAMING-tag, compared to 3' tagging with MS2-repeats. They show that transcription signal comes up sooner in the STREAMING-tag cells, conform to its 5' placement. The STREAMING-tagged cells, however, appear to show a bimodal distribution in activation. Is this reflecting true polII kinetics or is an artifact of the tagging? In addition, is the fraction of ON cells the same in both cell lines in this experiment? And how does the burst duration compare? Is this affected by splicing of the MS2 loops from the STREAMING-tag co-transcriptionally? How do these findings compare to published Nanog transcription dynamics? In addition, it appears that the coat protein expression levels in panel c and d differ: does this affect the ability to find MS2 spots above background? To conclude that STREAMING-tag is useful and does not affect dynamics of transcription, further characterization is required. In addition, in several places in the manuscript, proper quantification or controls are lacking:

4. Figure 1A shows optimization of the different inteins. It is unclear how the authors controlled for differences in transfection efficiency upon determining the fraction of

surviving cells.

5. Figure 1D shows two representative images of nuclear localization of the construct. How variable is expression between cells? How representative is this image? The authors should quantify this

6. In Figure 6, the distance between RF clusters and the gene is measured in ON and OFF state. What are the characteristics of these clusters, do RPB1, BRD4 or MED clusters differ in size, and how does this affect distance measurements? Many mouse embryonic stem cells are in G2 states, so two alleles could be distinguished. How does this affect interpretation of the data?

7. In Figure 6, panel f and g, the authors show 2 traces of 2D distance between RF clusters and the MCP signal. Such anecdotal traces are really not scientific proof, as single cells are stochastic and any example may occur. If the authors cannot get this experiment to work, I question the usefulness of is this system for such dynamic analysis.

8. Extended Data Fig3 shows overlap of the TetR label and MCP signal, and the Nanog locus using DNA-FISH. One example image is not sufficient here. How often do the DNA locus from DNA-FISH and tag overlap? Quantification of overlap in more cells would be beneficial.

Minor comments:

9. The authors decided to use the MS2 Sirius in their construct. I would like to see their choice motivated further, as to my knowledge the use of this type of loops is not common in the field.

10. The graphical representation of a hypothesis in both Fig. 6A, as well as 7d are not very clear. In case of the distant cluster representation, the cluster is surrounded by enhancers, but not in the case of a cluster in close proximity, why is this? What do the question marks represent? What are the two questions that are being depicted? It would be helpful to the reader to add a description of the model in the figure legend.

11. I would recommend making the code available upon publication.

Overall, the manuscript is conceptually interesting, but I would recommend more thorough experiments and analyses to show how well the system represents true transcription kinetics, before it is suitable for publication.

Reviewer #2 (Remarks to the Author):

The authors present a cleverly designed reporter for transcriptional activity and show evidence to demonstrate that the STREAMING-tag can reveal how transcriptional initiation is coordinated by relevant transcription factors and mediators on selected promoters. Comparisons are made with already published results using the MS2 system and the ROLEX system that help to corroborate the results reported here. However, as presented, the report lacks rigor and needs additional results and information regarding the analysis for its validation as discussed below:

1) As the reporter method relies on the affinities of proteins to RNA molecules, these interactions should be validated. That is, is the accumulation of fluorescent proteins such as MCP-RFP and TetR-mNG driven by their interactions with their RNA ligands or by some

other feature in the nucleus. Mutant proteins that do not bind their RNA ligands or RNAs that cannot bind the proteins should be used as negative controls. The control and experimental images and should be evaluated by a blinded analyst.

2) There is no discussion of image analysis. Was it blinded? This is particularly important for the time course studies. Were they analyzed as a series in time or were the time points scrambled prior to analysis? Was the analyst aware of the position in the time course of each image? Similar question apply to other image analysis results shown in this manuscript.

There are also some minor points that relate to the statistical analysis performed on the data.

1) On page 7 the authors report that they performed a pairwise T-test on the data to compare TetR(W43F) and TetR(WT). It is not clear on what basis is pairwise test is justified as it appears that these are all independently performed experiments and to pair individual experimental results for statistical analysis does not seem appropriate.

2) As well, the authors report accuracies to three significant figures, which are unreasonable in consideration of the methods employed and the reported errors in the estimates.

3) For each data set presented, the authors should state how many independent experiments were performed to obtain the data or, for data like cell images that are from one experiment, how many independent experiments of the same design were performed that provided similar data.

Reviewer #3 (Remarks to the Author):

Review: Ohishi et al "STREAMING-tag system reveals spatiotemporal relationships between transcriptional regulatory factors and transcriptional activity"

The authors developed an imaging system that allows to label a specific DNA locus and its nascent mRNAs. The novelty in this system lies in the addition of sequences that allow the elimination of the imaging module and the selection cassette by RNA splicing and protein splicing respectively. It also allows the incorporation of the tag downstream of the TSS to capture early transcriptional events such as initiation and pause release, as well as productive elongation. In addition, the manuscript presents new imaging tools that will be of tremendous interest to the community. These are CRISPR SNAP-tagged cell lines for pivotal transcriptional regulators (Rpb1, Brd4, Med19, Med22) and a genetically encoded mintbody for Pol II Ser5-P.

These tools are validated extensively with biochemical approaches, as well as quantitative imaging.

Parallel to technology development, the manuscript presents exciting biological results regarding the spatio-temporal control of Nanog transcriptional activity. They found that while Rbp1 and Brd4 clusters are associated with the transcriptionally active Nanog locus, MED19 or MED22 clusters are in proximity to the Nanog locus during periods of transcriptional quiescence as well as during active transcription.

The experiments described here constitute a technical tour-de-force that yield unique

insights into the dynamic regulation of gene transcription. The conclusions of this paper are of immediate interest to many in the field of transcription factors and gene expression. In particular, this study provides an important contribution to our understanding of how regulatory factors (TFs and GTFs) work within nuclear microenvironments to regulate transcription bursts. The main conclusions are well supported by the experimental data and I have only a few issues for the author's consideration.

Major comments:

-it's unclear whether the STREAMING-tag system reaches single molecule sensitivity for the detection of nascent mRNA. In general, with 24MS2 repeats, single molecule detection in live cells is challenging. The authors should discuss this point. In particular, a calibration of live imaging data with smFISH experiments could be helpful.

Related to this point, sentences such as 'In the STREAMING-tag system, MCP spots were anticipated to appear soon after the transcription of the Nanog start codon.' (p12 line 6) (or p11 line 22/23) should mention if the first nascent mRNA will be visualized with this system.

-assessing transcriptional activity in Nanog KI cells by smFISH: figure 2

Panel f is unclear: I suggest scoring the % of cells harboring: 1 active TS, 2 active TS or 0 active TS (assuming these cells are diploid in nature).

The sentence commenting on this part of the manuscript in the discussion is clearer than in the results section. (P12 line 32)

As the authors have counted the number of mature cytoplasmic mRNA in each condition, it would be interesting to know the distribution of TS intensities, expressed as 'single molecule equivalents', thus corresponding to absolute number of initiated Polymerase at the transcription site. A similar quantification of TS number and their levels (expressed in single molecule equivalents) could be performed for the other STREAMING-tag clones, Sox2 and/or Usp5.

-interpretation of MS2 live imaging data: could the authors clarify how they determined the 'ON' and 'OFF' states? In the absence of single molecule sensitivity and with a 5' tag, the time of appearance of the fluorescent transcription spot may not be the best proxy to determine the onset of the 'ON' state.

-quantification of locus mobilities is not sufficiently explained

Example: p8 line 7. Given the very good quality of the imaging, the authors could use quantitative approaches allowing to extract diffusion properties of a specific locus.

For example: DOI: 10.1038/s41467-021-26466-7

-In Figure 5, the authors record the time of appearance of transcription foci of Nanog-MS2 upon various drug treatments. Are these cells synchronized in terms of cell cycle?

Regardless of the answer to this point, the authors should discuss a potential effect of the cell cycle.

Minor comments:

-I found the introduction too detailed for the CTD part

-p3 line 35: cite Benabdallah et al., Mol Cell

-p4line3: I would reformulate the sentence to clarify what is meant by the 'OFF state' Maybe replace 'OFF state' by 'transcriptionally quiescent state.

-there are a large variety of abbreviations used, which causes some confusion during reading. It would be helpful to reduce the number of different abbreviations for the sake of clarity of understanding.

-what are the main differences between these 2 cell lines? What is the advantage of performing the experiments in these two biological contexts?

-Extended data fig2: the text above southern blots is too small to be legible

-p9 line1: unclear why the authors use a new cell line 'NM-G'. The authors could comment on the difference in timing of transcriptional reactivation after drug treatment between the two cell lines examined (2-3min vs 8-9min).

-p9 line 13: specify that the SNAP-tag generated lines are CRISPR lines, mention if the tag affects cell survival/physiological properties.

Point-by-point Responses to the Reviewers:

Manuscript number: NCOMMS-22-05929-T

Title: STREAMING-tag system reveals spatiotemporal relationships between transcriptional regulatory factors and transcriptional activity

We thank the reviewers for their advice and comments. This manuscript has benefited from these insightful suggestions. We have detailed our responses in black, with the reviewers' remarks in green. Changes and revisions to the main manuscript are highlighted in red, including the correction of typographical errors.

REVIEWER COMMENTS

Reviewer #1 (Remarks to the Author):

The manuscript of Ohishi et al. describes a novel system named STREAMING-tag that allows for labeling of DNA and RNA at a single locus in a single integration step. The system is an upgrade of their previous ROLEX system, allowing the study of a gene locus both when it is OFF (transcriptionally inactive) or ON (transcriptionally active). Using this system, the authors show that clusters of RNA Polymerase, Brd4 and Mediator close to the locus. Conceptually, such a system could be highly valuable for the community and has the potential to study spatiotemporal aspects of transcription. However, there are a number of important concerns that limit its usability in its current form. In addition, the manuscript frequently lacks proper quantification and could benefit from more thorough analysis.

My major concern is that the tag appears to interfere with transcription, the process being studied:

1. The authors show a clear reduction in expression levels at all integrated genes. Tagging always has the risk of interfering with function. If there is a major loss of function from tagging, as is the case here, then one should hesitate to use the system. Instead, the author just state the loss of expression, and do not appear too concerned. Although the authors claim that this effect mostly comes from post-transcriptional effects, I am currently not convinced by the provided evidence. In Figure 2, the number of Nanog transcripts is shown, as well as the fraction of cells showing a transcription site in the tagged and untagged allele. However, the untagged allele is also mutated and could be reduced as well, as there is no fully functional allele in the cell. For Sox2, and

perhaps for other genes, there is cellular feedback, where loss of function of one allele can affect both alleles. It is therefore important to also compare transcription of these cells to untagged wildtype cell lines? In addition, only burst frequency is analysed, but does the addition of the tag affect burst size (intensity of transcription site in smFISH)?

We analyzed the effect of STREAMING-tag knock-in on the transcriptional properties of the inserted genes. We performed smFISH in *Nanog*-, *Sox2*-, and *Usp5*-STREAMING-tag knock-in and WT cells using probes corresponding to the knock-in target genes. Based on the smFISH data, we quantified the number of mRNAs per cell and estimated the percentage of cells that showed bright smFISH spots in the nucleus (to indicate the bursting transcription sites) and the number of RNA molecules per transcription site as a proxy for burst size.

The results of smFISH analysis showed a slight decrease in RNA expression levels by STREAMING-tag knock-in for all genes (*Nanog*, 25.0%; *Sox2*, 9.0%; *Usp5*, 26.9%) (Fig. 2e, Supplementary Figs. 2e and 3e). The number of RNAs at the transcription site from the knock-in allele was slightly increased in *Nanog* and *Sox2* STREAMING-tag knock-in cells (7.3 vs 6.2 for *Nanog*; 4.9 vs 4.0 for *Sox2* on average), but did not change in the *Usp5* STREAMING-tag knock-in cells (Fig. 2f, Supplementary Figs. 2f and 3f). The percentage of transcriptionally active alleles in all cell lines was similar to that in WT cells (Fig. 2g, Supplementary Figs. 2g, and 3g). Considering the insertion of a significantly long cassette (5.5 kb), the decreased expression of the knock-in allele observed by western blotting and smFISH may be explained by the additional time required for elongation and splicing (Castillo-Davis et al., *Nat Genet*, 2002; Marais et al., *Genetics*, 2005; Swinburne et al., *Dev Cell*, 2008). In summary, STREAMING-tag knock-in reduced mRNA expression of target genes (down to 73%–91%) and tended to increase the number of RNAs retained in transcription sites (100%–123%), but the burst frequency was unaffected. We have added Fig. 2f, g, Supplementary Fig. 2 and 3 and revised the sentences related to these results accordingly (lines 163–186 and 1011–1016).

Reference:

1. Castillo-Davis, C. I., Mekhedov, S. L., Hartl, D. L., Koonin, E. V. & Kondrashov, F. A. Selection for short introns in highly expressed genes. *Nat Genet* **31**, 415–418 (2002).
2. Marais, G., Nouvellet, P., Keightley, P. D. & Charlesworth, B. Intron Size and Exon Evolution in *Drosophila*. *Genetics* **170**, 481–485 (2005).
3. Swinburne, I. A. & Silver, P. A. Intron Delays and Transcriptional Timing during Development. *Dev Cell* **14**, 324–330 (2008).

2. Moreover, the analysis in Fig 2 are performed in the absence of the TetR, and the authors show that TetR expression can majorly affect transcription in Figure 3. Although the authors optimize the TetR protein to reduce its affinity, they do not go on to show that transcription levels in their optimized system are similar with and without TetR(W43F) expression. The authors should directly compare the bursting kinetics (frequency and size) in cells without and without expression of TetR(W34F). In addition, they should check whether TetR(W43F) expression results in PolII stalling or premature termination, for example by Pol II-ChIP-seq.

We analyzed the effect of TetR(W43F)-mNG expression on transcription using smFISH, live imaging, and ChIP-qPCR. NLS-tagged mNG (NLS-mNG) and TetR(WT)-mNG were used as non-DNA-binding and strong DNA-binding protein controls, respectively. The number of *Nanog* mRNA detected by smFISH was similar in NLS-mNG and TetR(W43F)-mNG, but reduced in TetR(WT)-mNG (Fig. 3g). In living cells, the percentage of cells with MCP transcription spots and the fluorescent intensities of those spots were also similar in NLS-mNG and TetR(W43F)-mNG, whereas they were markedly reduced in TetR(WT)-mNG (Fig. 3h, i). ChIP-qPCR using an RNAPII-specific antibody showed that RNAPII occupancy on the STREAMING-tag was not affected by TetR(W43F)-mNG like TetR(WT)-mNG (Fig. 3j). These results support the hypothesis that TetR(W43F) does not affect STREAMING-tag transcription. We have added Fig. 3g-j and the sentence that explains this result (lines 222–234, 1051–1082).

3. In Figure 5 the authors show transcriptional dynamics quantification using STREAMING-tag, compared to 3' tagging with MS2-repeats. They show that transcription signal comes up sooner in the STREAMING-tag cells, conform to its 5' placement. The STREAMING-tagged cells, however, appear to show a bimodal distribution in activation. Is this reflecting true polII kinetics or is an artifact of the tagging? In addition, is the fraction of ON cells the same in both cell lines in this experiment? And how does the burst duration compare? Is this affected by splicing of the MS2 loops from the STREAMING-tag co-transcriptionally? How do these findings compare to published Nanog transcription dynamics? In addition, it appears that the coat protein expression levels in panel c and d differ: does this affect the ability to find MS2 spots above background? To conclude that STREAMING-tag is useful and does not affect dynamics of transcription, further characterization is required.

We have received similar comments from reviewers #2 and #3. Instead of comparing the STREAMING-tag with the previous 3' UTR MS2 by the DRB-release assay in the original manuscript, we have now included new data using a single cell line with a dual reporter system.

We established a cell line that harbors TSS-proximal STREAMING-tag and 3' UTR PP7 repeat at the *Nanog* gene. In this *Nanog*-STREAMING-PP7 cell line, the TSS-proximal region and 3' UTR of *Nanog* transcripts were monitored using MCP-RFP and PCP-HaloTag, respectively. The MCP-RFP reached a maximum intensity before the PCP-HaloTag. Cross-correlation analysis revealed that MCP-RFP preceded PCP-HaloTag for ~4 min (Fig. 5c-e). Thus, STREAMING-tag knocked-in near TSSs allows for a better quantification of transcriptional activity immediately after bursty transcription initiation. We have revised Fig. 5, added Supplementary Fig. 7, and revised the sentences related to these results (lines 308–322, 872–876, 1215–1231).

In addition, in several places in the manuscript, proper quantification or controls are lacking:
4. Figure 1A shows optimization of the different inteins. It is unclear how the authors controlled for differences in transfection efficiency upon determining the fraction of surviving cells.

We appreciate the reviewer's comments and have repeated the experiment in Fig. 1 with an evaluation of transfection efficiency. We performed flow cytometry analysis 2 days after transfection and confirmed that the percentage of mNG-positive cells did not differ between the samples (Fig. 1b). A fraction of transfected cells that were not used for flow cytometry analysis was subjected to analysis of cell survival rates in G418. CInt-I resulted in a higher percentage of surviving cells than PInt-I, as in the initial manuscript. We have revised Fig. 1b and the main text accordingly (lines 128–129 and 824–828).

5. Figure 1D shows two representative images of nuclear localization of the construct. How variable is expression between cells? How representative is this image? The authors should quantify this

We have repeated this analysis, provided new images, and quantified the data. Cells transfected with CInt-I and PInt-I constructs were grown in G418, fixed with 4% paraformaldehyde, stained with Hoechst 33342, and fluorescence images were acquired using a confocal microscope (Fig. 1d). Although a large variability in mNG expression levels was observed in the cell population, the fluorescence intensity was reproducibly higher in CInt-I than in PInt-1 (Fig. 1e). We have revised Fig. 1d, e, and f and the main text accordingly (lines 131–135 and 832–845).

6. In Figure 6, the distance between RF clusters and the gene is measured in ON and OFF state. What are the characteristics of these clusters, do RPB1, BRD4 or MED clusters differ in size, and

how does this affect distance measurements? Many mouse embryonic stem cells are in G2 states, so two alleles could be distinguished. How does this affect interpretation of the data?

The size and intensity of the RF clusters were measured. The cluster sizes of all RF nearest neighbors of mTetR spots were in a 500–600 nm range regardless of the transcription state, which simplifies the interpretation (Supplementary Fig. 8f, g). The intensities of the RPB1 and BRD4 clusters were higher in the ON state, whereas those of the MED19 and MED22 clusters were constant (Supplementary Fig. 8e). The distance measurements were not affected by the intensity because the coordinates of the RF clusters were determined by the local maxima. We have added Supplementary Fig. 8e-g and a sentence that explains this result (lines 351–356 and 1375–1380).

The percentage of cells in the S/G2 phase is ~60%–70% (Waisman et al., *Sci Rep*, 2019) in mESCs. As mentioned by the reviewer, genomic regions after DNA replication are often observed as “doublet” spots (Stanyte et al., *J Cell Biol*, 2018). During data acquisition for the original manuscript, we mainly selected cells that showed “singlet” mTetR spots for imaging. We apologize for not mentioning this in the original version. We have now double checked our key data to exclude cells showing “doublet” spots and ~1.5% of cells that were found to be “doublet”. After removing these doublet cells, we reanalyzed the data presented in Fig. 6 and 7 and Supplementary Fig. 8. In the revised manuscript, we have clarified that we analyzed “singlet” cells that are probably G1 and S phase cells before replication of the *Nanog* gene. We have added a relevant description in the Results section and discussed future research related to the cell cycle in the Discussion (lines 267–270).

References:

1. Waisman, A. *et al.* Cell cycle dynamics of mouse embryonic stem cells in the ground state and during transition to formative pluripotency. *Sci Rep* **9**, 8051 (2019).
2. Stanyte, R. *et al.* Dynamics of sister chromatid resolution during cell cycle progression. *J Cell Biol* **217**, 1985–2004 (2018).

7. In Figure 6, panel f and g, the authors show 2 traces of 2D distance between RF clusters and the MCP signal. Such anecdotal traces are really not scientific proof, as single cells are stochastic and any example may occur. If the authors cannot get this experiment to work, I question the usefulness of is this system for such dynamic analysis.

Thank you for your comments. We have removed Fig. 6f and g and the relevant descriptions from the revised manuscript, as we considered the analysis to be insufficient.

8. Extended Data Fig3 shows overlap of the TetR label and MCP signal, and the Nanog locus using DNA-FISH. One example image is not sufficient here. How often do the DNA locus from DNA-FISH and tag overlap? Quantification of overlap in more cells would be beneficial.

Due to the low number of data taken, we performed the experiment again. In this experiment, cells were fixed with 4% paraformaldehyde, and mTetR-mNG, MCP-RFP, and Hoechst 33342 fluorescence images were acquired. After DNA-FISH using *Nanog* probe, the same cells were imaged. We used this sequential imaging method because mTetR-mNG and MCP-RFP fluorescence cannot tolerate the harsh DNA-FISH conditions. However, DNA-FISH procedure often causes relatively large deformation of the cell nuclei and therefore, the images before and after FISH cannot be merged in a straight way. We corrected the deformation of the nucleus using a Fiji plugin Fijiyama (Fernandez et al., *Bioinformatics*, 2020) based on Hoechst images. Even after the deformation correction, the two images before and after FISH were not matched perfectly. Therefore, mTetR-mNG spots did not always overlap with one of two DNA-FISH spots that are likely to represent the *Nanog* loci (Supplementary Fig. 4a). Nevertheless, 41% of mTetR-mNG had a *Nanog* FISH spot within 500 nm (median distance to the nearest mTetR: 584 nm) (Supplementary Fig. 4b, c). The distance between mTetR and another (not the nearest) DNA-FISH spots was much higher (median 6.8 μm). Thus, mTetR spots label the *Nanog* gene vicinity region. We have added Supplementary Fig. 4a-c, and revised sentences related to this result (lines 214–221 and 1115–1121).

Reference:

1. Fernandez, R. & Moisy, C. Fijiyama: a registration tool for 3D multimodal time-lapse imaging. *Bioinformatics* **37**, 1482–1484 (2020).

Minor comments:

9. The authors decided to use the MS2 Sirius in their construct. I would like to see their choice motivated further, as to my knowledge the use of this type of loops is not common in the field.

Sirius MS2 is a reduced sequence similarity between each MS2 stem-loop (Ma et al., *Nat Methods*, 2018). It has been reported that the fluorescence intensity of spots is higher when 8 \times Sirius MS2

is used as compared to that in the conventional 14× MS2 repeats in the dCas9 system. Therefore, we adopted the Sirius MS2 for transcription reporter. We have added an explanation to the main text (lines 102–109).

Reference:

1. Ma, H. *et al.* CRISPR-Sirius: RNA Scaffolds for Signal Amplification in Genome Imaging. *Nat Methods* **15**, 928–931 (2018).

10. The graphical representation of a hypothesis in both Fig. 6A, as well as 7d are not very clear. In case of the distant cluster representation, the cluster is surrounded by enhancers, but not in the case of a cluster in close proximity, why is this? What do the question marks represent? What are the two questions that are being depicted? It would be helpful to the reader to add a description of the model in the figure legend.

We apologize for these inappropriate figures. We have revised Fig. 6a and 7d accordingly.

11. I would recommend making the code available upon publication.

We will make Python code, which was used for detecting mTetR spots and RF clusters, and to analyze the distance between mTetR spots and RF clusters, available on GitHub. We have added the URL to the GitHub site (lines 1399–1402).

Overall, the manuscript is conceptually interesting, but I would recommend more thorough experiments and analyses to show how well the system represents true transcription kinetics, before it is suitable for publication.

We thank the reviewer for several important comments on how the STREAMING-tag represents endogenous transcription kinetics. By adding data that show that STREAMING-tag knock-in and mTetR expression minimally affects the initiation of bursty transcription, we believe the manuscript is now more technically sound and provides important information for the organization of transcription.

Reviewer #2 (Remarks to the Author):

The authors present a cleverly designed reporter for transcriptional activity and show evidence to demonstrate that the STREAMING-tag can reveal how transcriptional initiation is coordinated by relevant transcription factors and mediators on selected promoters. Comparisons are made with already published results using the MS2 system and the ROLEX system that help to corroborate the results reported here. However, as presented, the report lacks rigor and needs additional results and information regarding the analysis for its validation as discussed below:

1) As the reporter method relies on the affinities of proteins to RNA molecules, these interactions should be validated. That is, is the accumulation of fluorescent proteins such as MCP-RFP and TetR-mNG driven by their interactions with their RNA ligands or by some other feature in the nucleus. Mutant proteins that do not bind their RNA ligands or RNAs that cannot bind the proteins should be used as negative controls. The control and experimental images and should be evaluated by a blinded analyst.

We performed two additional experiments to verify that the spots of MCP-RFP represent STREAMING-tag transcripts containing Sirius MS2. The first experiment used WT cells that did not express MS2 as the negative control. In the second experiment, MCP mutants that are known to bind less to MS2 were used. These results support the view that transcripts containing Sirius MS2 are MCP-RFP spots. The details follow.

First, we compared the number of MCP-RFP spots in *Nanog* STREAMING knock-in and WT mESCs. To identify MCP-RFP spots based on their intensity, we used the local maxima method with Trackpy. The number of spots per nucleus was determined by setting a threshold for fold enrichment of 2×, 3×, 4×, and 5× over the nuclear background intensity. Although the number of spots varied depending on the threshold level, they were much higher in *Nanog* STREAMING knock-in cells than in WT cells (Supplementary Fig. 4d, e). When the threshold to define a spot was set 3-fold over the average nuclear intensity, 63.5% of *Nanog* STREAMING knock-in cells showed MCP-RFP spots, whereas 94.1% of WT cells showed no spots (Supplementary Fig. 4d, e). This indicates that most MCP-RFP spots represent knock-in *Nanog* RNA containing Sirius MS2.

We next examined whether MCP S47R and R49H mutants, which have reduced MS2 binding ability (Peabody et al., *Nucleic Acids Res*, 1996; Duan et al., *Nucleic Acids Res*, 2021), impaired their accumulation at Sirius MS2. The fluorescence intensities of MCP(S47R)-RFP and MCP(S47R, R49H)-RFP on mTetR-mNG spots were on average 3.0- and 5.3-fold lower, respectively, than those of WT MCP-RFP (Supplementary Fig. 4f, g). Thus, the spot appearance of MCP-RFP depended on its binding to Sirius MS2 RNA.

Regarding the mTetR spot, DNA-FISH analysis confirmed that the mTetR spot was associated with one of the *Nanog* loci (please see our response to the 8th comment to Reviewer #1). We believe this indicates that mTetR specifically recognizes the STREAMING tag region.

In our image analysis, the same parameters were used throughout the samples to be compared; therefore, the results were not analyst-dependent.

We have added Supplementary Fig. 4d-g and a paragraph explaining these results (lines 236–267 and 1123–1146).

Reference:

1. Peabody, D. S. & Lim, F. Complementation of RNA Binding Site Mutations in MS2 Coat Protein Heterodimers. *Nucleic Acids Res* **24**, 2352–2359 (1996).
2. Duan, N., Arroyo, M., Deng, W., Cardoso, M. C. & Leonhardt, H. Visualization and characterization of RNA–protein interactions in living cells. *Nucleic Acids Res* **49**, gkab614- (2021).

2) There is no discussion of image analysis. Was it blinded? This is particularly important for the time course studies. Were they analyzed as a series in time or were the time points scrambled prior to analysis? Was the analyst aware of the position in the time course of each image? Similar question apply to other image analysis results shown in this manuscript.

Determining the time point when an MCP spot appears (original Fig. 5) and the choice of an example of intensity fluctuation in time (original Fig. 6f, g) were indeed analyst-dependent. We have removed these figures from the revised manuscript.

The revised manuscript includes several image analyses. Most analyses were analyst-independent. In analyst-dependent cases, we systematically changed the parameters and validated their robustness. The details follow.

1. Measuring the intensity in the whole nucleus (Fig. 1d, e, Fig. 3h, Supplementary Fig. 4e, g)
We selected the nucleus using cellpose (Stringer et al., *Nat Methods*, 2021) based on Hoechst staining (Fig. 1d, e) or the fluorescence of reporters like MCP-RFP (Fig. 3h, Supplementary Fig. 4e, g), which is essentially analyst-independent.
2. Counting the number of spots (Fig. 2e, 3g, Supplementary Fig. 2e, 3e, 4e)
For smFISH (Fig. 2e, 3g, Supplementary Fig. 2e, 3e), we used Big-FISH (Imbert et al., *Rna*, 2022), which is essentially analyst-independent. For counting the number of MCP-RFP spots (Supplementary Fig. 4e), we first used Trackpy to detect spots and then further selected brighter

spots relative to nuclear background by thresholding. Although selecting the thresholding levels was analyst-dependent, we systematically analyzed by altering different threshold levels (Supplementary Fig. 4e).

3. Measuring the distance between spots (between spots in different colors and between time points) (Fig. 5d, 6c, 7b)

The Trackpy local maxima method was used to detect spots. The same parameter settings were used for all samples for comparison, including negative controls. The center of mass in a spot was automatically determined using Trackpy; therefore, the distances between spots in time-lapse movies and multicolor channels were calculated in an analyst-independent manner.

4. Measuring the intensity (Fig. 5, 6, 7, Supplementary Fig. 8)

The intensities of individual spots or clusters (in 6-pixel diameter) were measured using the Trackpy software. Most signals in a single spot were covered in a 6-pixel diameter circle (see Fig. 6b and 7a) as the full-width at half maxima of the spots was ~500–600 nm, corresponding to ~4 pixels (130 nm/pixel) (Supplementary Fig. 8g). We apologize for not describing this in the original manuscript.

Based on the MCP-RFP spot intensity with respect to the average intensity of the 19×19 pixel area and the distance between mTetR-mNG and the nearest MCP-RFP spots, we classified the ON and OFF states. In the original manuscript, we classified a spot in which the MCP-RFP intensity was more than 2-fold over the background. This threshold was determined after plotting a histogram of relative MCP-RFP intensity in the MCP-RFP spot region (6 pixel diameter) centered within 3 pixels of the mTetR-mNG spot. We initially considered setting the threshold at 2-fold as sensible as a peak was found below 2-fold. However, we understood that this was marginal and therefore performed a systematic analysis by altering the threshold level from 1 to 6 (Supplementary Fig. 8d). The result remained essentially the same when the threshold was varied from 2 to 5.5, indicating the robustness of the analysis. We retained the 2-fold threshold data in Fig. 6 and 7, and the entire data are provided in Supplementary Fig. 8d.

Instead of the original Fig. 5, in which we compared the STREAMING-tag with the previous 3' UTR MS2 by the DRB-release assay, we now included new data using a single cell line with a dual reporter system. For time-lapse imaging, we employed cross-correlation analysis (Fig. 5e), which is not analyst-dependent.

We have described the above in the Methods section.

Reference:

1. Stringer, C., Wang, T., Michaelos, M. & Pachitariu, M. Cellpose: a generalist algorithm for cellular segmentation. *Nat Methods* 18, 100–106 (2021).

2. Imbert, A. *et al.* FISH-quant v2: a scalable and modular tool for smFISH image analysis. *Rna* 28, rna.079073.121 (2022).

There are also some minor points that relate to the statistical analysis performed on the data.

1) On page 7 the authors report that they performed a pairwise T-test on the data to compare TetR(W43F) and TetR(WT). It is not clear on what basis is pairwise test is justified as it appears that these are all independently performed experiments and to pair individual experimental results for statistical analysis does not seem appropriate.

We have removed this statement because it was not appropriate to test here.

2) As well, the authors report accuracies to three significant figures, which are unreasonable in consideration of the methods employed and the reported errors in the estimates.

Overall, we have made changes to two significant figures.

3) For each data set presented, the authors should state how many independent experiments were performed to obtain the data or, for data like cell images that are from one experiment, how many independent experiments of the same design were performed that provided similar data.

We have now included the number of independent experiments and cells/spots per experiment. Because the data in the original Figs. 6 and 7 were in fact from a single set of experiments, we repeated the experiment and prepared new figures in the revised version (Fig. 6 and 7). As a result, the distance between mTetR spots and the nearest MED22 clusters was significantly different (in contrast to the previous version). We have revised the text taking into the account of the new data. The main conclusions were the same.

Reviewer #3 (Remarks to the Author):

Review: Ohishi et al "STREAMING-tag system reveals spatiotemporal relationships between transcriptional regulatory factors and transcriptional activity"

The authors developed an imaging system that allows to label a specific DNA locus and its

nascent mRNAs. The novelty in this system lies in the addition of sequences that allow the elimination of the imaging module and the selection cassette by RNA splicing and protein splicing respectively. It also allows the incorporation of the tag downstream of the TSS to capture early transcriptional events such as initiation and pause release, as well as productive elongation. In addition, the manuscript presents new imaging tools that will be of tremendous interest to the community. These are CRISPR SNAP-tagged cell lines for pivotal transcriptional regulators (Rpb1, Brd4, Med19, Med22) and a genetically encoded mintbody for Pol II Ser5-P. These tools are validated extensively with biochemical approaches, as well as quantitative imaging.

Parallel to technology development, the manuscript presents exciting biological results regarding the spatio-temporal control of Nanog transcriptional activity. They found that while Rbp1 and Brd4 clusters are associated with the transcriptionally active Nanog locus, MED19 or MED22 clusters are in proximity to the Nanog locus during periods of transcriptional quiescence as well as during active transcription.

The experiments described here constitute a technical tour-de-force that yield unique insights into the dynamic regulation of gene transcription. The conclusions of this paper are of immediate interest to many in the field of transcription factors and gene expression. In particular, this study provides an important contribution to our understanding of how regulatory factors (TFs and GTFs) work within nuclear microenvironments to regulate transcription bursts. The main conclusions are well supported by the experimental data and I have only a few issues for the author's consideration.

Major comments:

1. -it's unclear whether the STREAMING-tag system reaches single molecule sensitivity for the detection of nascent mRNA. In general, with 24MS2 repeats, single molecule detection in live cells is challenging. The authors should discuss this point. In particular, a calibration of live imaging data with smFISH experiments could be helpful.

We estimated RNA detection sensitivity using the STREAMING-tag system. The smFISH results showed that an average of 7.3 RNAs were present per transcription site in the STREAMING-tag knock-in allele (Fig. 2f). From the live imaging data, the relative intensity of MCP-RFP at the mTetR-mNG spot was ~7-fold higher than the nuclear background intensity (Supplementary Fig. 4g). This suggests that the MCP-RFP intensity for a single RNA molecule might be in a similar

range to that of the nuclear background. When the threshold of MCP-RFP signal was set to 3-fold above the background, 63.5% of cells showed MCP-RFP spots (Supplementary Fig. 4e). This number is comparable to the percentage of single alleles with transcripts by smFISH (~60%; 50% with biallelic expression, plus a half of 20% single allelic expression; Fig. 2g). Since our smFISH analysis defines transcription sites as RNA clusters of three or more molecules detected in the cell nucleus, it is reasonable that the percentage of cells with MCP-RFP spots whose relative intensity values are more than three on live imaging is equivalent to that of smFISH. Therefore, the detection of a single RNA molecule is not realistic under the experimental conditions used in this study. If multiple RNAs are transcribed simultaneously from an allele, the MCP spot can be detected and determined to be in the ON state. Therefore, to avoid misunderstanding, we have added a sentence defining the ON state in this study as the state in which the STREAMING-tag knock-in gene is transcribed by multiple RNAPIIs (lines 250–267).

2. Related to this point, sentences such as ‘In the STREAMING-tag system, MCP spots were anticipated to appear soon after the transcription of the Nanog start codon.’ (p12 line 6) (or p11 line 22/23) should mention if the first nascent mRNA will be visualized with this system.

As described above, detecting the first single RNA molecule is difficult. Instead, we detected a burst of transcription with more than one transcript at a transcription site. Therefore, we changed the text to say “transcription burst” or “bursty transcription.”

3. -assessing transcriptional activity in Nanog KI cells by smFISH: figure 2

Panel f is unclear: I suggest scoring the % of cells harboring: 1 active TS, 2 active TS or 0 active TS (assuming these cells are diploid in nature).

The sentence commenting on this part of the manuscript in the discussion is clearer than in the results section. (P12 line 32)

As the authors have counted the number of mature cytoplasmic mRNA in each condition, it would be interesting to know the distribution of TS intensities, expressed as ‘single molecule equivalents’, thus corresponding to absolute number of initiated Polymerase at the transcription site. A similar quantification of TS number and their levels (expressed in single molecule equivalents) could be performed for the other STREAMING-tag clones, Sox2 and/or Usp5.

According to the comments by reviewers #1 and #3, the part of smFISH was extensively rewritten by including (1) the number of transcripts at transcription sites estimated from smFISH

signals, (2) the percentages of cells showing 2, 1, and 0 transcription sites in the WT cells as well as *Nanog* STREAMING-tag knock-in cells, and (3) the analysis of both *Sox2* and *Usp5*.

We analyzed the effect of STREAMING-tag knock-in on the transcriptional properties of the inserted genes. We performed smFISH in *Nanog*-, *Sox2*-, and *Usp5*-STREAMING-tag knock-in and WT cells, using probes corresponding to the knock-in target genes. Based on the smFISH data, we quantified the number of mRNAs per cell and estimated the percentage of cells that showed bright smFISH spots in the nucleus (to indicate the bursting transcription sites) and the number of RNA molecules per transcription site as a proxy for burst size.

The results of smFISH analysis showed a slight decrease in RNA expression levels by STREAMING-tag knock-in for all genes (*Nanog*, 25.0%; *Sox2*, 9.0%; *Usp5*, 26.9%) (Fig. 2e, Supplementary Figs. 2e and 3e). The number of RNAs at the transcription site from knock-in allele was slightly increased in *Nanog* and *Sox2* STREAMING-tag knock-in cells (7.3 vs 6.2 for *Nanog*; 4.9 vs 4.0 for *Sox2* on average), but was not changed in *Usp5* STREAMING-tag knock-in cells (Fig. 2f, Supplementary Figs. 2f and 3f). The percentage of transcriptionally active alleles in both cell lines was similar to that in WT cells (Fig. 2g and Supplementary Figs. 2g and 3g). Considering the insertion of a significantly long cassette (5.5 kb), the decreased expression of the knock-in allele observed by western blotting and smFISH may be explained by the additional time required for elongation and splicing (Castillo-Davis et al., *Nat Genet*, 2002; Marais et al., *Genetics*, 2005; Swinburne et al., *Dev Cell*, 2008). In summary, STREAMING-tag knock-in reduced mRNA expression of target genes (down to 73%–91%) and tended to increase the number of RNAs retained in transcription sites (100%–123%), but the burst frequency was unaffected. We have added Fig. 2f, g, Supplementary Fig. 2 and 3, and revised the sentences related to these results accordingly (lines 163–186 and 1011–1016).

Reference:

1. Castillo-Davis, C. I., Mekhedov, S. L., Hartl, D. L., Koonin, E. V. & Kondrashov, F. A. Selection for short introns in highly expressed genes. *Nat Genet* **31**, 415–418 (2002).
2. Marais, G., Nouvellet, P., Keightley, P. D. & Charlesworth, B. Intron Size and Exon Evolution in *Drosophila*. *Genetics* **170**, 481–485 (2005).
3. Swinburne, I. A. & Silver, P. A. Intron Delays and Transcriptional Timing during Development. *Dev Cell* **14**, 324–330 (2008).

4. *-interpretation of MS2 live imaging data: could the authors clarify how they determined the 'ON' and 'OFF' states? In the absence of single molecule sensitivity and with a 5' tag, the time*

of appearance of the fluorescent transcription spot may not be the best proxy to determine the onset of the 'ON' state.

We have received similar comments from reviewers #1 and #2, and have incorporated the same in our previous replies.

Could the authors clarify how they determined the 'ON' and 'OFF' states?

The intensities of individual spots or clusters (in 6-pixel diameter) were measured using Trackpy. Most signals in a single spot were covered in a 6-pixel diameter circle (see Fig. 6b and 7a) as the full-width at half maxima of the spots was ~500–600 nm, corresponding to ~4 pixels (130 nm/pixel) (Supplementary Fig. 8g). We apologize for not describing this in the original manuscript.

Based on the MCP-RFP spot intensity with respect to the average intensity of the 19×19 pixel area and the distance between mTetR-mNG and the nearest MCP-RFP spots, we classified the ON and OFF states. In the original manuscript, we classified a spot in which the MCP-RFP intensity was more than 2-fold over the background. This threshold was determined after plotting a histogram of relative MCP-RFP intensity in the MCP-RFP spot region (6 pixel diameter) centered within 3 pixels of the mTetR-mNG spot. We initially considered setting the threshold at 2-fold as sensible because a peak was found below 2-fold. However, we understood that this was marginal and therefore performed a systematic analysis by altering the threshold level from 1 to 6 (Supplementary Fig. 8d). The result remained essentially the same when the threshold was varied from 2 to 5.5, indicating the robustness of the analysis. We retained the 2-fold threshold data in the main Figures, while the whole data are provided in Supplementary Fig. 8d.

In the absence of single molecule sensitivity and with a 5' tag, the time of appearance of the fluorescent transcription spot may not be the best proxy to determine the onset of the 'ON' state.

The timing of MCP spot appearance in the STREAMING-tag was compared with the previous 3' UTR MS2 using the DRB-release assay, as shown in the original Fig. 5. We have deleted this data. Instead, we included new data using a single-cell line with a dual reporter system. We established a cell line that harbors TSS-proximal STREAMING-tag and 3' UTR PP7 repeat in the *Nanog* gene. In this *Nanog*-STREAMING-PP7 cell line, the TSS-proximal region and 3' UTR of *Nanog* transcripts were monitored using MCP-RFP and PCP-HaloTag, respectively. The MCP-RFP reached a maximum intensity before PCP-HaloTag reached a maximum ~4 min (Fig. 5c-e). Thus, STREAMING-tag knocked-in near TSSs allows for a better quantification of transcriptional activity immediately after bursty transcription initiation. We revised Fig. 5, added

Supplementary Fig. 7, and revised the sentences related to these results (lines 308–322, 873–875, 1215–1231).

5. *-quantification of locus mobilities is not sufficiently explained*

Example: p8 line 7. Given the very good quality of the imaging, the authors could use quantitative approaches allowing to extract diffusion properties of a specific locus.

For example: DOI: 10.1038/s41467-021-26466-7

We measured mobility over a relatively long term (for several minutes) in the original manuscript. We have now included other short-term data in (a few seconds) to determine the diffusion coefficient. The trend of increased mobility of genomic loci in the OFF state was also observed in short-term measurements, which was consistent with previous reports (Nagashima et al., *J Cell Biology*, 2019). We have described in detail how we measured the mobilities (lines 280–286). We have added Supplementary Fig. 6 and revised the main text accordingly (lines 299–306, 1194–1212).

Reference:

1. Nagashima, R. *et al.* Single nucleosome imaging reveals loose genome chromatin networks via active RNA polymerase II. *J Cell Biology* **218**, 1511–1530 (2019).

6. *-In Figure5, the authors record the time of appearance of transcription foci of Nanog-MS2 upon various drug treatments. Are these cells synchronized in terms of cell cycle? Regardless of the answer to this point, the authors should discuss a potential effect of the cell cycle.*

The original Fig. 5 has been deleted and replaced with the dual-tag experimental data. The percentage of cells in S/G2 phase is ~60%–70% (Waisman et al., *Sci Rep*, 2019) in mESCs. Genomic regions after DNA replication are often observed as “doublet” spots (Stanyte et al., *J Cell Biol*, 2018). During the data acquisition for the original manuscript, we had mainly selected cells that showed “singlet” mTetR spots for imaging; we apologize not mentioning this in the original version. We have now double checked our key data to exclude cells showing “doublet” spots and ~1.5% of cells that were found to be “doublet.” After removing those doublet cells, we reanalyzed the data presented in Fig. 6 and 7 and Supplementary Fig. 8. In the revised manuscript, we have mentioned that we analyzed “singlet” cells that are probably G1 and S phase cells before

replication of the *Nanog* gene. We have added relevant description in the Results section and discussed the future research related to the cell cycle in Discussion (lines 267–270, 447–454).

References:

1. Waisman, A. *et al.* Cell cycle dynamics of mouse embryonic stem cells in the ground state and during transition to formative pluripotency. *Sci Rep* **9**, 8051 (2019).
2. Stanyte, R. *et al.* Dynamics of sister chromatid resolution during cell cycle progression. *J Cell Biol* **217**, 1985–2004 (2018).

Minor comments:

7. *-I found the introduction too detailed for the CTD part*

We have simplified the CTD part (lines 51–53).

8. *-p3 line 35: cite Benabdallah et al., Mol Cell*

We have cited the paper (line 72).

9. *-p4line3: I would reformulate the sentence to clarify what is meant by the 'OFF state' Maybe replace 'OFF state' by 'transcriptionally quiescent state.'*

We have now explained the ON and OFF states. We think using ON and OFF in figures and the main text is easier than transcriptionally “active” and “quiescent” states (lines 56–57).

10. *-there are a large variety of abbreviations used, which causes some confusion during reading. It would be helpful to reduce the number of different abbreviations for the sake of clarity of understanding.*

We have reduced abbreviations as much as possible.

11. *-what are the main differences between these 2 cell lines? What is the advantage of performing the experiments in these two biological contexts?*

I think the reviewer means “NSt” and “NSt-GR,” or “NSt” and “NM-G.” We apologize for the confusion. “NSt-GR” was an “NSt”-derived line with mTetR-mNG and MCP-RFP expression. However, we have now avoided using these abbreviations in the main text, and have just described “*Nanog* STREAMING-tag knock-in cells” for “NSt” and “*Nanog* STREAMING-tag knock-in cells expressing mTetR-mNG and MCP-RFP” for “NSt-GR”. We deleted the data using “NM-G,” which would improve clarity.

12. -Extended data fig2: the text above southern blots is too small to be legible

We have enlarged the text of the Southern blot in Supplementary Figs. 2 and 3.

13. -p9 line1: unclear why the authors use a new cell line 'NM-G'. The authors could comment on the difference in timing of transcriptional reactivation after drug treatment between the two cell lines examined (2-3min vs 8-9min).

We apologize for the lack of a detailed explanation. We hypothesized that we could measure the difference in time from transcription start to transcription spot detection by comparing NM-G, a cell line with an MS2 sequence inserted in the 3' UTR, with *Nanog* STREAMING-tag knock-in cells with STREAMING-tag knock-in near the TSS. However, because of the issue of defining the appearance time point, the data in the original Fig. 5 using these cells were deleted, and then NM-G was no longer used in the revised manuscript.

14. -p9 line 13: specify that the SNAP-tag generated lines are CRISPR lines, mention if the tag affects cell survival/physiological properties.

Thank you for this suggestion. We have specified in the main text that these cells are SNAPtag knock-in cells using CRISPR-Cas9 gene editing (line 333-334). We have also noted in the Methods section that these SNAPtag knock-in cells have not been affected in proliferation (lines 947–949).

REVIEWERS' COMMENTS

Reviewer #1 (Remarks to the Author):

The revised manuscript of Ohishi et al. is much improved compared to the original submission. My comments have been adequately addressed. The manuscript now includes quantifications and thorough analysis that support the claims. There are just three minor points that require addressing:

-The authors observe decreased expression of the knock-in allele observed by western blotting and smFISH and they explain this by 'the additional time required for elongation and splicing' (line 187-189). If this were the case, this additional time needs to be very substantial to result in much less transcripts and proteins. A much more likely result is that the addition of the tag results alterations in the mRNA splicing/processing/export and translation process which can affect the mRNA and protein degradation rates. Please include a discussion with alternative options in the manuscript.

-In the discussion (line 459-460), the authors claim that the addition of the STREAMING tag 'does not appear to significantly affect the initiation of bursty transcription'. This is a bit confusing because 'initiation of bursty transcription' is not a term used in literature, and some bursting parameters (such as intensity) are affected by their tag. It is better to use the correct nomenclature and to be specific, by stating that the 'active fraction' or 'burst frequency' is not affected.

-The authors should describe in the methods section (from line 1344) how they image the clusters, or at the very least, refer to a previous publication. Which microscopy system was used, what imaging settings, etc? Only the analysis and culture conditions are described now.

If these three points are adjusted, I recommend publication of this article.

Reviewer #2 (Remarks to the Author):

The authors have done a thorough job in responding to the reviewers both with revisions in the presentation and in the added (and deleted) data to support their conclusions. With some further corrections of spelling errors (such as in lines 720 and 789 in the revised version) I believe that this manuscript is acceptable for publication.

Reviewer #3 (Remarks to the Author):

The manuscript is significantly improved. The additional smFISH data are extremely useful. overall I am happy that the authors have addressed my all of my concerns. congratulations!

Point-by-point Responses to the Reviewers:

Manuscript number: NCOMMS-22-05929B

Title: STREAMING-tag system reveals spatiotemporal relationships between transcriptional regulatory factors and transcriptional activity

We thank the reviewers for their advice and comments. We have detailed our responses in black, with the reviewers' remarks in green. Changes and revisions to the main manuscript are highlighted in red.

REVIEWER COMMENTS

Reviewer #1 (Remarks to the Author):

The revised manuscript of Ohishi et al. is much improved compared to the original submission. My comments have been adequately addressed. The manuscript now includes quantifications and thorough analysis that support the claims. There are just three minor points that require addressing:

1. The authors observe decreased expression of the knock-in allele observed by western blotting and smFISH and they explain this by 'the additional time required for elongation and splicing' (line 187-189). If this were the case, this additional time needs to be very substations to result in much less transcripts and proteins. A much more likely result is that the addition of the tag results alterations in the mRNA splicing/processing/export and translation process which can affect the mRNA and protein degradation rates. Please include a discussion with alternative options in the manuscript.

As Reviewer #1 pointed out, the tag could be causing the decrease in RNA expression by altering the mRNA splicing, processing, and export processes, thereby affecting the rate of mRNA degradation. The relevant sentences have been altered (lines 187–189, 465–467).

2. In the discussion (line 459-460), the authors claim that the addition of the STREAMING tag 'does not appear to significantly affect the initiation of bursty transcription'. This is a bit confusing because 'initiation of bursty transcription' is not a term used in literature, and some bursting

parameters (such as intensity) are affected by their tag. It is better to use the correct nomenclature and to be specific, by stating that the 'active fraction' or 'burst frequency' is not affected.

We did not use the term “burst frequency” in our original manuscript. Therefore, we have revised “initiation of bursty transcription” to "fraction of cells with transcription spots, representing the transcription burst frequency^{19,50}." (lines 464–465)

3. The authors should describe in the methods section (from line 1344) how they image the clusters, or at the very least, refer to a previous publication. Which microscopy system was used, what imaging settings, etc? Only the analysis and culture conditions are described now.

We have added information about the imaging conditions as well as references to the section that describes the microscope settings used in detail. (Lines 1068–1074).

Reviewer #2 (Remarks to the Author):

The authors have done a thorough job in responding to the reviewers both with revisions in the presentation and in the added (and deleted) data to support their conclusions. With some further corrections of spelling errors (such as in lines 720 and 789 in the revised version) I believe that this manuscript is acceptable for publication.

We have revised typos as pointed out by Reviewer #2.

Reviewer #3 (Remarks to the Author):

The manuscript is significantly improved. The additional smFISH data are extremely useful. overall I am happy that the authors have addressed my all of my concerns. congratulations!